# DISCS: A Benchmark for Discrete Sampling

**Katayoon Goshvadi**
Google Deepmind

**Haoran Sun**
Georgia Tech

**Xingchao Liu**
UT Austin

**Azade Nova**
Google Deepmind

**Ruqi Zhang**
Purdue University

**Will Grathwohl**
Google Deepmind

**Dale Schuurmans**
Google Deepmind

**Hanjun Dai**
Google Deepmind

## Abstract

Sampling in discrete spaces, with critical applications in simulation and optimization, has recently been boosted by significant advances in gradient-based approaches that exploit modern accelerators like GPUs. However, two key challenges are hindering further advancement in research on discrete sampling. First, since there is no consensus on experimental settings and evaluation setups, the empirical results in different research papers are often not comparable. Second, implementing samplers and target distributions often requires a nontrivial amount of effort in terms of calibration and parallelism. To tackle these challenges, we propose *DISCS* (DISCrete Sampling), a tailored package and benchmark that supports unified and efficient experiment implementation and evaluations for discrete sampling in three types of tasks: sampling from classical graphical models and energy based generative models, and sampling for solving combinatorial optimization. Throughout the comprehensive evaluations in *DISCS*, we gained new insights into scalability, design principles for proposal distributions, and lessons for adaptive sampling design. *DISCS* efficiently implements representative discrete samplers in existing research works as baselines and offers a simple interface that researchers can conveniently add new discrete samplers and directly compare their performance with the benchmark result in a calibrated setup.

## 1  Introduction

Sampling in discrete spaces has been an important problem for decades in physics (Edwards & Anderson, 1975; Baumgärtner et al., 2012), statistics (Robert & Casella, 2013; Carpenter et al., 2017), and computer science (LeCun et al., 2006; Wang & Cho, 2019). Since sampling from a target probability distribution $\pi(x) \propto \exp(-f(x))$ defined on a discrete space $\mathcal{X}$ is typically intractable, one usually resorts to Markov chain Monte Carlo (MCMC) methods (Metropolis et al., 1953; Hastings, 1970). However, except for a few algorithms such as Swendsen-Wang for the Ising model (Swendsen & Wang, 1987) and Hamze-Freitas for hierarchical models (Hamze & de Freitas, 2012), which exploit the special structure of the underlying problem, sampling in a general discrete space has primarily relied on Gibbs sampling, which exhibits notoriously poor efficiency in high dimensional spaces.

Recently, a family of locally balanced MCMC samplers for discrete spaces (Zanella, 2020; Grathwohl et al., 2021; Sun et al., 2021; Zhang et al., 2022), using ratio informed proposal distributions, $\frac{\pi(y)}{\pi(x)}$, have significantly improved sampling efficiency by exploiting modern accelerators like GPUs and TPUs. From the perspective of gradient flow on the Wasserstein manifold of distributions, Gibbs sampling is simply a coordinate descent algorithm, whereas locally balanced samplers perform as full gradient descent (Sun et al., 2022a). Despite the advances in locally balanced samplers, a quantitative benchmark is still missing. As a result, the empirical results in different research papers may not be comparable. One important reason is that there is no consensus on the experimental

setting. Particularly, the initialization of energy based generative models, random seeds used in graphical models, and the protocol of hyper-parameter tuning all have a significant impact on samplers performance. Under this circumstance, there is a critical need for a unified benchmark to advance the research in discrete sampling.

There are two key challenges that seriously hinder the appearance of such a benchmark. First, a sampler may perform well in one target distribution while poorly in another one. To thoroughly examine the performance of a sampler, a qualified benchmark needs to collect a set of representative distributions that covers the potential applications of discrete samplers. Second, the evaluation of discrete samplers is complicated. Although the commonly used metric Effective Sample Size (ESS) (Vehtari et al., 2021) can effectively reflect the efficiency of a sampler in Monte Carlo integration or Bayesian inference, it is not very informative in scenarios when the sampler guides the search in combinatorial optimization problems or performs as a decoder in deep generative models.

To address the two challenges, we propose *DISCS*, a tailored benchmark for discrete sampling. In particular, *DISCS* consists of three groups of tasks: sampling from classical graphical models, sampling for solving combinatorial optimization problems, and sampling from deep energy based models (EBMs). These tasks cover both the topics of simulation and optimization, and also target distributions, ranging from hand-designed graphical models to learned deep EBMs. For each task, we collect the representative problems from both synthetic and real-world applications, for example, graph partitioning for distributed computing and language model for text generation. We carefully design the evaluation metrics in *DISCS*. In sampling from classical graphical models tasks, *DISCS* uses ESS as a standard. In sampling for solving combinatorial optimization tasks, *DISCS* runs simulated annealing (Kirkpatrick et al., 1983) with multiple chains and reports the average of the best results in each chain. In sampling from energy based generative models, *DISCS* employs domain specific scores to measure the sample quality.

*DISCS* offers a convenient interface for researchers to implement new discrete samplers, without worrying about parallelism, experiment loop, and evaluation. *DISCS* can efficiently sweep over different tasks and experiments configurations in parallel, making it easy to reproduce the benchmark results of this paper. Also, *DISCS* implements existing discrete samplers including random walk Metropolis (Metropolis et al., 1953), block Gibbs, Hamming Ball sampler (Titsias & Yau, 2017), Locally Balanced (Zanella, 2020), Gibbs with Gradients (Grathwohl et al., 2021), Path Auxiliary Sampler (Sun et al., 2021), Discrete Metropolis Adjusted Langevin Algorithm (Zhang et al., 2022), Discrete Langevin Monte Carlo (Sun et al., 2022a), and is actively maintaining to add new samplers. Researchers can directly compare the results with state-of-the-art methods.

*DISCS* can also provide insights on the existing open questions in the space of discrete sampling. In our experiments, we observe an interesting phenomenon that the locally balanced weight function $g(t) = \sqrt{t}$ outperforms $g(t) = \frac{t}{t+1}$ when Ising model has a temperature higher than the critical temperature and $g(t) = \frac{t}{t+1}$ outperforms $g(t) = \sqrt{t}$ when the temperature is lower than the critical temperature. We further observe similar phenomenon in our experiments with more complicated deep energy generative model where $g(t) = \frac{t}{t+1}$ outperforms $g(t) = \sqrt{t}$ on overparameterized neural network with low temperature and sharp landscape. There have been numerous studies on how to select the locally balanced function for a locally balanced sampler (Zanella, 2020; Sansone, 2022), but the answer still remains open. We hope the observations in this paper can provide some insight on this question.

We wrap the *DISCS* package as a JAX library to facilitate the research in discrete sampling. The library is open sourced at DISCS [1]. The dataset used in our benchmark experiments can be accessed at DISCS DATA [2] The paper is organized as follows:

- In section 2, we provide an overview of related work on different tasks for discrete sampling and recent advances in discrete samplers.
- In section 3, we formulate the discrete sampling problem.
- In section 4, we introduce the discrete sampling tasks and evaluation metrics in *DISCS*. We also present several results with interesting insights and observations.
- In section 5, we discuss the contribution and limitations of *DISCS*.

---

[1] code base found at `https://github.com/google-research/discs`

[2] data set found at `https://drive.google.com/drive/u/1/folders/1nEppxuUJj8bsV9Prc946LN_buo30AnDx`

We provide comprehensive results of our benchmark and studies with further details on the experimental setups, evaluation metrics, mathematical formulations, and the data set used in the Appendix A.

## 2 Related Work

Discrete sampling has been widely used to study the physical picture of spin glasses (Hukushima & Nemoto, 1996; Katzgraber et al., 2001), solve combinatorial optimization via simulated annealing (Kirkpatrick et al., 1983), and for training or decoding deep energy based generative models (Wang & Cho, 2019; Du et al., 2020; Dai et al., 2020b). However, these tasks primarily depend on Gibbs sampling, which could be very slow in high dimensional space.

Since the seminal work Zanella (2020), the recent years have witnessed significant progresses for discrete sampling in both theory and practice. Zanella (2020) introduces the locally balanced proposal $q(x, y) \propto g(\frac{\pi(y)}{\pi(x)})$, where $y \in N(X)$ restricted within a small neighborhood of $x$ and $g(\cdot) : \mathbb{R}_+ \to \mathbb{R}_+$ satisfying $g(a) = ag(\frac{1}{a})$, and prove it is asymptotically optimal. In the following works, PAS (Sun et al., 2021) and DMALA (Zhang et al., 2022) generalize locally balanced proposal to large neighborhoods by introducing an auxiliary path and mimicking the diffusion process, respectively. Inspired by these locally balanced samplers, Sun et al. (2022a) generalize the Langevin dynamics in continuous space to *discrete Langevin dynamics* (DLD) in discrete space as a continuous time Markov chain $\frac{d}{dh}\mathbb{P}(X^{t+h} = y|X^t = x) = g(\frac{\pi(y)}{\pi(x)})$, and show that previous locally balanced samplers are simulations of DLD with different discretization strategies. In the view of Wasserstein gradient flow, the Gibbs sampling can be seen as coordinate descent and DLD gives a full gradient descent. Hence, locally balanced samplers induced from DLD provide a principled framework to utilize modern accelerators like GPUs and TPUs to accelerate discrete sampling. Besides the discretization of DLD, another crucial part to design a locally balanced sampler is estimating the probability ratio $\frac{\pi(y)}{\pi(x)}$. Grathwohl et al. (2021) proposes to used gradient approximation $\frac{\pi(y)}{\pi(x)} \approx \exp(-\langle\nabla f(x), y - x\rangle)$ and obtains good performance on various classical models and deep energy based models. When the Hessian is available, Rhodes & Gutmann (2022); Sun et al. (2023a) use second order approximation via Gaussian integral trick (Hubbard, 1959) to further improve the sampling efficiency on skewed target distributions. When the gradient is not available, Xiang et al. (2023) use zero order approximation via Newton's series.

Besides designing the sampler, Sun et al. (2022b) proves that when tuning path length in PAS (Sun et al., 2021), the optimal efficiency is obtained when the average acceptance rate is 0.574, and design an adaptive tuning algorithm for PAS. Sansone (2022) learn locally balanced weight function for locally balanced proposal, but how to select the weight function in a principled manner is still unclear.

## 3 Formulation for Sampling in Discrete Space

The sampling in discrete space can be formulated as the following problem: in a finite discrete space $\mathcal{X}$, we have an energy function $f(\cdot) : \mathcal{X} \to \mathbb{R}$. We consider a target distribution

$$\pi(x) = \frac{\exp(-\beta f(x))}{Z}, \quad Z = \sum_{z \in \mathcal{X}} \exp(-\beta f(z)), \tag{1}$$

where $\beta$ is the inverse temperature. When the normalizer $Z$ is intractable, people usually resort to Markov chain Monte Carlo (MCMC) to sample from the target distribution $\pi$. Metropolis-Hastings (M-H) (Metropolis et al., 1953; Hastings, 1970) is a commonly used general purpose MCMC algorithm. Specifically, given a current state $x^{(t)}$, the M-H algorithm proposes a candidate state $y$ from a proposal distribution $q(x^{(t)}, y)$. Then, with probability

$$\min\left\{1, \frac{\pi(y)q(y, x^{(t)})}{\pi(x^{(t)})q(x^{(t)}, y)}\right\}, \tag{2}$$

the proposed state is accepted and $x^{(t+1)} = y$; otherwise, $x^{(t+1)} = x^{(t)}$. In this way, the detailed balance condition is satisfied and the M-H sampler generates a Markov chain $x^{(0)}, x^{(1)}, ...$ that has $\pi$ as its stationary distribution.

# 4 Benchmark for Sampling in Discrete Space

The recent development of locally balanced samplers that use the ratio $\frac{\pi(y)}{\pi(x)}$ to guide the proposal distribution $q(x, \cdot)$ has significantly improved the sampling efficiency in discrete space. However, there is no consensus for many experimental settings. As a result, the empirical results in different research papers may not be directly comparable. Under this circumstance, we propose *DISCS* as a benchmark for general purpose samplers in discrete space. In section 4.1, we introduce implemented sampling methods as the baselines in *DISCS* and make some remarks on how we present the results. *DISCS* implements both the classical discrete samplers and the recent developed locally balanced samplers. In the following sections, we introduce the tasks considered in *DISCS* as follows: sampling from classical graphical models (section 4.2), sampling for solving combinatorial optimization problems (section 4.3), and sampling from deep energy based generative models (section 4.4). We also describe how the discrete samplers are evaluated on these tasks and report several results of them with some insights. We leave the remaining comprehensive results and experimentation and report them in detail in Appendix A .

## 4.1 Baselines

We include both classical discrete samplers and state-of-the-art locally balanced samplers in recent research papers as baselines in our benchmark. Specifically, *DISCS* implements

1. random walk Metropolis (RWM) (Metropolis et al., 1953).
2. block Gibbs (BG), where BG-<a> denotes using block Gibbs with block size $a$.
3. Hamming Ball Sampler (HB) (Titsias & Yau, 2017), where HB-<a>- denotes using block size $a$ and Hamming ball size $b$.
4. Gibbs with Gradients (GWG) (Grathwohl et al., 2021), a locally balanced sampler that uses gradient to approximate the probability ratio. For binary distribution, GWG has a scaling factor $L$ to determine how many sites to flip per step.
5. Path Auxiliary Sampler (PAS) (Sun et al., 2021), a locally balanced sampler that has a scaling factor $L$ to determine the path length.
6. Discrete Metropolis Adjusted Langevin Algorithm (DMALA)(Zhang et al., 2022), a locally balanced sampler that has a scaling factor $\alpha$ to determine the step size.
7. Discrete Langevin Monte Carlo (DLMC) (Sun et al., 2022a), a locally balanced sampler that has a scaling factor $\tau$ to determine the simulation time of DLD. DLMC has multiple choices for its numerical solver to approximate the transition matrix. *DISCS* considers the two versions used in the original paper, DLMC that uses an interpolation, and DLMCf that uses Euler's forward method.

Note that in the depicted plots in this paper, each of the previously described samplers is represented with a distinct and unique color of its own. We present samplers RWM, BG, HB, GWG, PAS, DMALA, DLMC, DLMCf as the colors green, yellow, blue, red, brown, purple, pink and grey.

**Remark: weight function**    *DISCS* offers a range of locally balanced functions, including $g(t) = \sqrt{t}$, $g(t) = \frac{t}{t+1}$, $g(t) = 1 \wedge t$, and $g(t) = 1 \vee t$. All the locally balanced samplers have the flexibility to select from these locally balanced functions. $g(t) = \sqrt{t}$ and $g(t) = \frac{t}{t+1}$ are the two most commonly used weight functions which we also rely on for our experiments. We use <sampler>-<func> to refer to the type of the weight function for the locally balanced sampler. In cases that the weight function is not reported, we use $\sqrt{t}$ by default.

**Remark: scaling**    Since the scaling of the proposal distribution in RWM, PAS, DMALA, and DLMC are tunable, we consider two versions: one with adaptive tuning and another with binary search tuning to ensure fair comparison. Sun et al. (2022b, 2023b) propose an adaptive tuning algorithm for PAS and DLMC when the target distribution is factorized. In practice, we find that they also apply well to other locally balanced samplers and for more general target distributions. Hence, in this paper, we use the adaptive tuning algorithm by default to tune the scaling for locally balanced samplers. In the several exceptions where the adaptive algorithm does not apply, we will use <sampler-name>-noA to indicate the results from binary search tuning.

## 4.2 Sampling from Classical Graphical Models

This section covers the classical graphical models frequently employed in physics and statistics, including Bernoulli Models, Ising Models (Ising, 1924), and Factorial Hidden Markov Models (Ghahramani & Jordan, 1995). The graphical models have large configuration flexibility, for example, the number of discrete variables, the number of categories for each discrete variable, and the temperature of the model. The performances of different samplers can heavily depend on these configurations. *DISCS* provides tools to automatically sweep over hundreds of configurations with one click. Following the common practice in Monte Carlo integration or Bayesian inference, *DISCS* uses the Effective Sample Size (ESS) to evaluate the efficiency of each sampler and reports the ESS normalized by the number of calling energy functions and by the running time. Below, we present several Ising Model results as illustrative examples. We report more results and details of our experiments and ESS calculation in the Appendix A.1, A.6.1. More specifically, we report the performance results of different samplers on Bernoulli Model 7, Categorical Model 8, Ising Model 9, Potts Model 10 and FHMM 11. We study the effect of number of discrete variables (sample dimension), the number of categories for each discrete variable, weight function for locally balanced samplers, and the temperature of the models (smoothness/sharpness).

The Ising Model is defined on a 2D grid, where the state space $\mathcal{X} = \{-1, 1\}^{p \times p}$ represents the spins on all nodes. For each state $x \in \mathcal{X}$, the energy function is defined as:

$$f(x) = -\sum_{i,j} J_{ij} x_i x_j - \sum_i h_i x_i \qquad (3)$$

where $J_{ij}$ is the internal interaction and the $h_i$ is the external field. In the main text, we report the results using the configuration from Zanella (2020). Specifically, $J_{ij} = 0.5$, $h_i = \mu_i + \sigma_i$, where $\sigma_i \sim \text{Uniform}(-1.5, 1.5)$ and $\mu_i = 0.5$ if node $i$ is located in a circle has the same center as the 2D grid and radius $\frac{p}{2\sqrt{2}}$, else $-0.5$. We consider the target distribution $\pi(x) \propto \exp(-\beta f(x))$, where $\beta$ is the inverse temperature. Using *DISCS*, one can easily investigate the influence of the number of discrete variables (sample dimension). In Figure 1, one can see that the classical samplers, RWM, BG, HB, have a significant decrease in ESS when the model dimension increases, while the locally balanced samplers are less affected as the ratio information $\frac{\pi(y)}{\pi(x)}$ effectively guides the proposal distribution. The overall trends basically follows the prediction from Sun et al. (2022b) that the ESS is $O(d^{-1})$ for RWM and $O(d^{-\frac{1}{3}})$ for PAS.

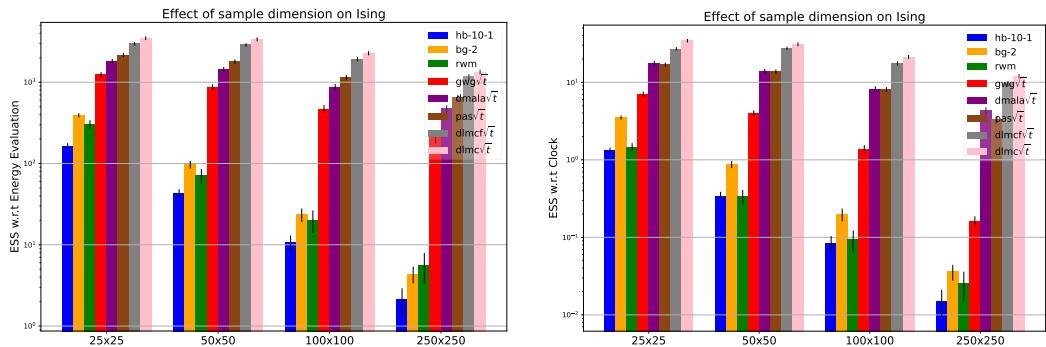

Figure 1: Results of Ising model with different dimensions.

Through *DISCS*, researchers can easily define various experiments, configure tasks, evaluate sampler performance and gain invalubale insights on open questions. As an example, we experiment with sampling from Ising Models with a range of temperatures and sample dimensions. In Figure 2, we experiment with sampling from Ising Models with inverse temperatures from 0.1607 to 0.7607 for both sample dimension of $50 \times 50$ and $100 \times 100$. We consider Ising Model without external field: $h_i \equiv 0$ and $J_{ij} \equiv 1$ as we know the critical temperature for this configuration is $\frac{2}{\log(1+\sqrt{2})}$. This gives us the critical point for inverse temperature as $\beta = 0.4407$. From the results in Figure 2, we can see that

- The Ising model is harder to sample from when the inverse temperature $\beta$ is closer to the critical point, which is consistent with the theory in statistical physics.

- When the inverse temperature $\beta$ is lower than the critical point, using weight function $g(t) = \sqrt{t}$ gives larger ESS; When the inverse temperature is larger than the critical point, using weight function $g(t) = \frac{t}{t+1}$ consistently obtains larger ESS.

The second observation implies that one should use ratio function $\frac{t}{t+1}$ for target distributions with a sharp landscapes. We will revisit this conclusion in Table 2. We report more results on the effect of sample dimension in Appendix (7, 8, 9, 10).

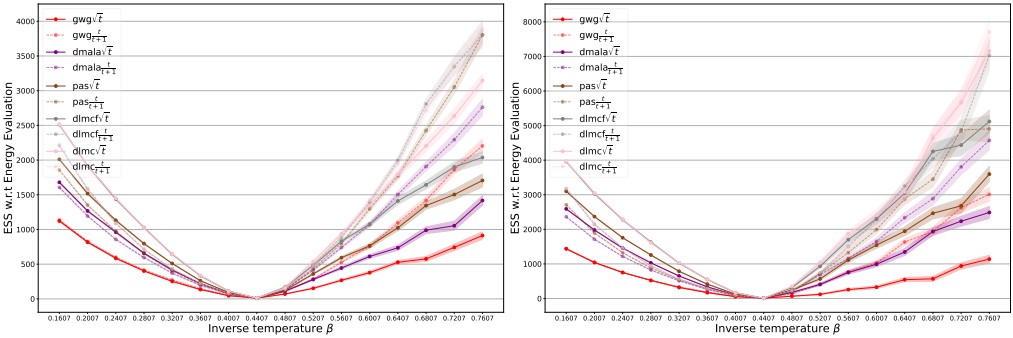

Figure 2: Performance of locally balanced samplers with different types of weight functions v.s temperature on: (left) $50 \times 50$ Ising model, (right) $100 \times 100$ Ising model.

The categorical version of Ising model is Potts model, where each site of a state $x_i$ has values in a symmetry group, instead of $\{-1, 1\}$. For simplicity, we denote the symmetry group as a set of one hot vectors $\mathcal{C} = \{e_1, ..., e_c\}$ with $h_i \in \mathbb{R}^C, J_{ij} \in \mathbb{R}^{C \times C}$. In this way, the energy function becomes:

$$f(x) = -\sum_{i,j} x_i^\top J_{ij} x_j - \sum_i \langle h_i, x_i \rangle \tag{4}$$

In Figure 3, one can see the sampling efficiency is very robust with respect to the number of categories, with the exception of the classical HB and BG samplers, which demonstrate a decline in their sampling performance. The result for BG-2 on Potts model with 256 categories is omitted as it takes over 100 hours. We report more results on the effect of number of categories in Appendix (8, 10, 11).

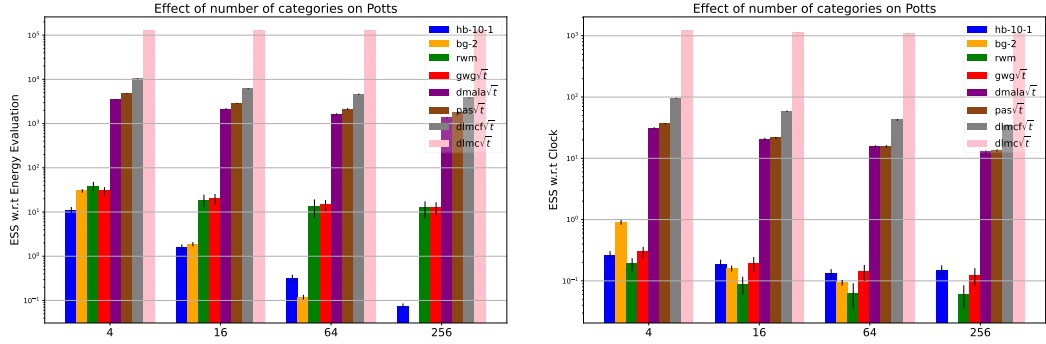

Figure 3: Results of Potts models with different number of categories.

## 4.3 Sampling for Solving Combinatorial Optimiazation

Combinatorial optimization is a core challenge in domains like logistics, supply chain management, and hardware design, and has been a fundamental problem of study in computer science for decades. Combining with simulated annealing Kirkpatrick et al. (1983), the discrete sampling algorithm is a powerful tool to solve combinatorial optimization problems (Sun et al., 2023b). In expectation, a sampler with a faster mixing rate can find better solutions. Hence, the second type of task is sampling for solving combinatorial optimization problems. Currently, *DISCS* covers four problems:

Maximum Independent Set (MIS), Max Clique, MaxCut, and Balanced Graph Partition. Without loss of generality, we consider combinatorial optimization that admits the following form:

$$\min_{x \in \mathcal{C} = \{0,1,\ldots,C-1\}^d} a(x), \quad \text{s.t.} \quad b(x) = 0 \tag{5}$$

For ease of exposition, we assume $b(x) \geq 0, \forall x \in \mathcal{C}$, but otherwise do not limit the form of $a$ and $b$. To convert the optimization problem to a sampling problem, we first rewrite the constrained optimization into a penalty form via a penalty coefficient $\lambda$, then treat this as an energy function for an EBM. In particular, the energy function takes the form:

$$f(x) = a(x) + \lambda \cdot b(x) \tag{6}$$

Then, we define the probability of $x$ at inverse temperature $\beta$ by:

$$p_\beta(x) \propto \exp(-\beta f(x)) \tag{7}$$

A naive approach to this problem would be directly sampling from $p_{\beta \to \infty}(x)$, but such a distribution is highly nonsmooth and unsuitable for MCMC methods. Instead, following classical simulated annealing, we define a sequence of distributions parameterized by a sequence of decaying temperatures:

$$\mathcal{P} = [p_{\beta_0}(x), p_{\beta_1}(x), \ldots, p_{\beta_T}(x)] \tag{8}$$

where the sequence $\beta_0 < \beta_1 < \ldots < \beta_T \to \infty$ converges to a large enough value as $T$ increases. Below, we present the problem formulation and the energy functions used for MaxCut and MIS problems. We present several results of the samplers solution for these combinatorial optimization problems as illustrative examples in the main text and report the remaining results in the Appendix A.2.

**MaxCut** The objective of MaxCut problem is to find a cut on a graph $G = (V, E)$ that partitions the graph nodes into two complementary sets $V = V_1 \cup V_2$, such that the number of edges in $E$ between $V_1$ and $V_2$ is as large as possible. MaxCut is an unconstrained problem, which makes its formulation relatively simple. We can set $\mathcal{C} = \{0, 1\}$ such that $x_i = 0$ represents $i \in V_1$ and $x_i = 1$ means $x_i \in V_2$. Then we can write $a(x) = -x^\top A x, b(x) \equiv 0$, where $A$ is the adjacency matrix of the graph $G$.

We apply the same simulated annealing temperature scheduling set up for all the samplers and compare the samplers performances against each other. In the MaxCut problem, we compute the ratio of our solution against the optimal solution found by Gurobi, running for one hour (Dai et al., 2020a). The results presented at Figure 4 show the cut ratio throughout the chain generation over the number of M-H steps and the running time (s). The legends are sorted according to the most optimal solution each sampler finds. One can see that the PAS leads the results. Also, locally balanced samplers significantly outperform the traditional samplers, especially when the graph size increases.

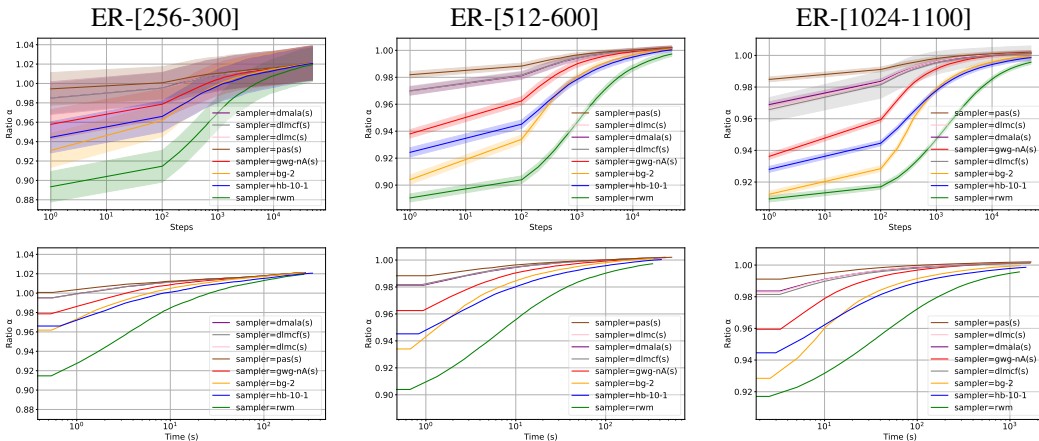

Figure 4: Results for MaxCut on ER graphs. The ratio is computed by dividing the optimal cut size obtained from running Gurobi for 1 hour. (top) ratio with respect to the number of M-H steps, (bottom) ratio with respect to running time.

**MIS** On a graph $G = (V, E)$, an independent set $S \subset V$ means that for any $i, j \in S$, $(i, j) \notin E$. We can set $\mathcal{C} = \{0, 1\}$ such that $x_i = 0$ means $i \notin S$ and $x_i = 1$ means $i \in S$. Then we can write $a(x) = -\sum_{i \in V} x_i$ and $b(x) = \sum_{(i,j) \in E} x_i x_j$. For the penalty coefficient $\lambda$, we follow Sun et al. (2022c) to select $\lambda = 1.0001$ being a value slightly larger than 1. We run all samplers on five groups of small ER graphs with 700 to 800 nodes, each group has 128 graphs with densities varying 0.05, 0.10, 0.15, 0.20, and 0.25. We also run all samplers on 16 large ER graphs with 9000 to 11000 nodes. For each configuration, we run 32 chains with the same running time and report the average of the best results found by each chain in Table 1. One can easily see that PAS obtains the best result.

Table 1: Results for MIS on ER graphs. The set found by the sampling algorithm is not necessarily an independent set, we report a lower bound: set size - # pair of adjacent nodes in the set.

| Sampler | ER[700-800] | | | | | ER[9000-11000] |
|---|---|---|---|---|---|---|
| | 0.05 | 0.10 | 0.15 | 0.20 | 0.25 | 0.15 |
| HB-10-1 | 100.374 | 58.750 | 41.812 | 32.344 | 26.469 | 277.149 |
| BG-2 | 102.468 | 60.000 | 42.820 | 32.250 | 27.312 | 316.170 |
| RWM | 97.186 | 56.249 | 40.429 | 31.219 | 25.594 | -555.674 |
| GWG-nA | 104.812 | 62.125 | 44.383 | **34.812** | 28.187 | 367.310 |
| DMALA | 104.750 | 62.031 | 44.195 | 34.375 | 28.031 | 357.058 |
| PAS | **105.062** | **62.250** | **44.570** | 34.719 | **28.500** | **377.123** |
| DLMCf | 104.450 | 62.219 | 44.078 | 34.469 | 28.125 | 354.121 |
| DLMC | 104.844 | 62.187 | 44.273 | 34.500 | 28.281 | 355.058 |

## 4.4 Sampling from Energy Based Generative Models

The discrete samplers can also play the role of decoder in generative models. In particular, given a dataset $\mathcal{D} = \{X_i\}_{i=1}^N$ sampled from the target distribution $\pi$, one can train an energy function $f_\theta(\cdot)$, such that the energy based model $\pi_\theta(\cdot) \propto \exp(-f_\theta(\cdot))$ fits the dataset $\mathcal{D}$. *DISCS* provides multiple checkpoints for the energy function trained on real-world image or language datasets. Researchers can easily evaluate their samplers after loading the learned energy function. We provide further experimental details and mathematical formulations at Appendix A.3.

For the models that are relatively simple, for example, Restricted Boltzmann Machine (RBM) trained on MNIST (LeCun, 1998) and fashion-MNIST (Xiao et al., 2017b), one can continue using ESS as the metric. In Figure 5, we evaluate the samplers on RBMs trained on MNIST with 25 and 200 hidden variables. One can see that DLMC has the best performance. We further report the results of samplers on categorical RBM trained on fashion-MNIST dataset at Appendix 16. For more complicated deep energy based models, a sampler may fail to mix within reasonable steps. In this case, ESS is not a good metric. To address this problem, *DISCS* provides multiple alternative measurements, including snapshots and domain specific scores.

**Snapshots** After loading the checkpoint of energy based generative models, *DISCS* can generate snapshots of the sampling chains. For example, in Figure 6, we display the snapshots of sampling on a deep residual network trained on MNIST data (Sun et al., 2021) and on pretrained language model BERT [3]. One can see that locally balanced samplers generate samples with higher qualities, and can typically visit multiple modalities in the distribution. We report further results on deep residual network trained on Omniglot and Caltech dataset at 17.

**Domain Specific Scores** In many deep generative tasks, the goal is to efficiently sample high-quality samples, instead of mixing in the learned energy based models. In this scenario, domain specific scores that directly evaluate the sample qualities are better choices. For example, *DISCS* provides text filling task based on pre-trained language models like BERT (Wang & Cho, 2019; Devlin et al., 2018). Following the settings in prior work (Zhang et al., 2022), we randomly samples 20 sentences from TBC (Zhu et al., 2015) and WiKiText-103 (Merity et al., 2016) and mask four words in each sentence (Donahue et al., 2020) resulting in the dataset provided at DISCS DATA. For each masked sentence, we sample 25 sentences by generating 25 chains with length of 50, following the target probability distribution provided by BERT, and then selecting the last sample of the chain. As a common practice

---

[3] loading the checkpoint from `https://huggingface.co/bert-base-uncased`.

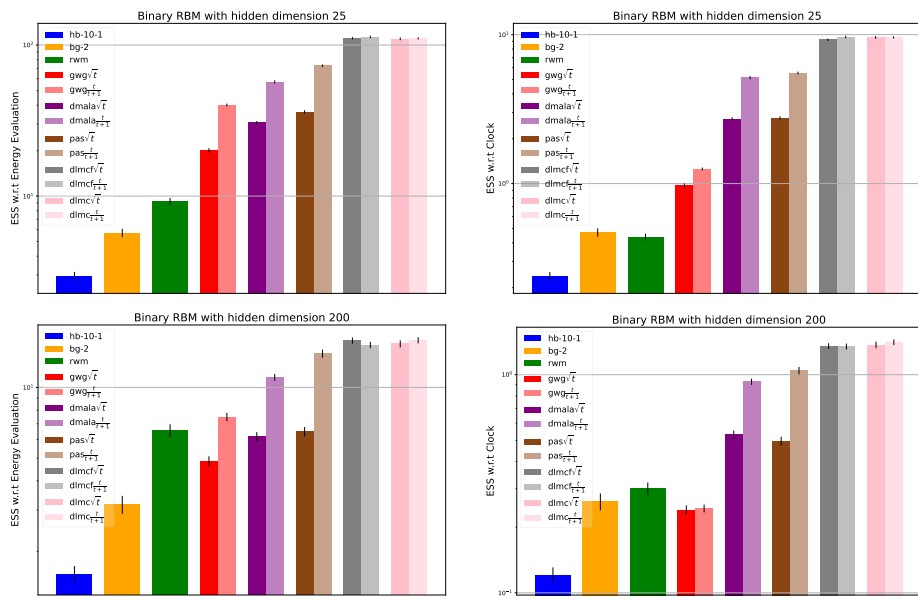

Figure 5: Results on RBMs trained on MNIST dataset. (top) RBM with 25 binary hidden variables, (bottom) RBM with 200 binary hidden variables.

in non-auto-regressive text generation, we select the top-5 sentences with the highest likelihood out of 25 sentences to avoid low-quality generation (Gu et al., 2017; Zhou et al., 2019).

We evaluate the generated samples in terms of diversity and quality. For diversity, we use self-BLEU (Zhu et al., 2018) and the number of unique n-grams (Wang & Cho, 2019) to measure the difference between the generated sentences. For quality, we measure the BLEU score (Papineni et al., 2002) between the generated texts and the original dataset, which is the combination of TBC and WikiText-103. We report the quantitative results in Table 2. We do not have the results for HB and BG as they are computationally infeasible for this task with 30k+ tokens. In this task, the locally balanced sampler still outperforms RWM. Also, one can notice that the weight function $\frac{t}{t+1}$ significantly outperforms $\sqrt{t}$ on diversity metrics and reaches comparable results on the quality. The reason is that the overparameterized neural network is a low temperature system with sharp landscape. This phenomenon is consistent with the results in Figure 2. We provide further results for the non-adaptive cases with binary search fine tuning in Appendix A.3.3.

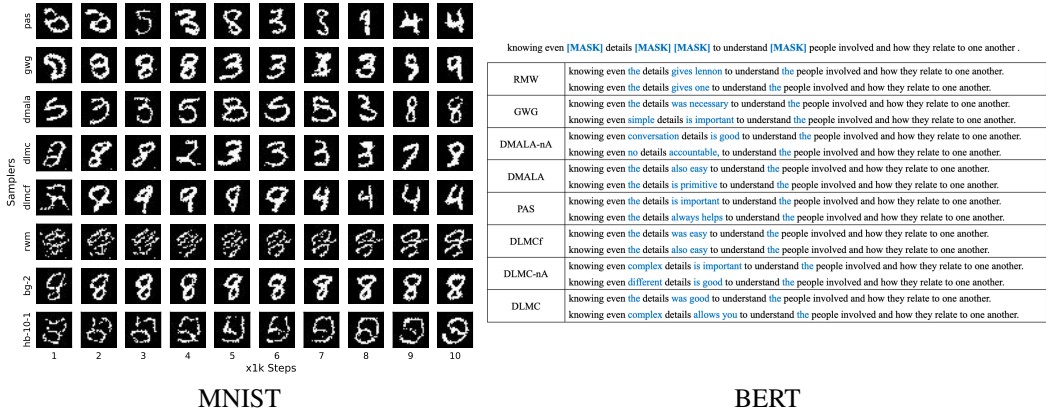

MNIST                                        BERT

Figure 6: Snapshots of energy based generative models: (left) snapshots for every 1k steps on MNIST ResNet, (right) snapshots for text filling task on BERT in Table 2

Table 2: Quantative results on text infilling. The reference text for computing the Corpus BLEU is the combination of WT103 and TBC.

| Methods | Self-BLEU (↓) | Unique $n$-grams (%) (↑) | | | | | | Corpus BLEU (↑) |
|---|---|---|---|---|---|---|---|---|
| | | Self | | WT103 | | TBC | | |
| | | $n=2$ | $n=3$ | $n=2$ | $n=3$ | $n=2$ | $n=3$ | |
| RWM | 92.41 | 6.26 | 9.10 | 18.97 | 26.73 | 19.33 | 26.67 | 16.24 |
| GWG$\sqrt{t}$ | 85.93 | 11.22 | 17.14 | 23.16 | 35.56 | 23.58 | 35.56 | 16.75 |
| DMALA$\sqrt{t}$ | 85.88 | 11.58 | 17.14 | 22.07 | 34.08 | 23.22 | 34.15 | **17.06** |
| PAS$\sqrt{t}$ | 85.39 | 11.37 | 17.60 | 22.61 | 35.53 | 23.65 | 35.47 | 16.57 |
| DLMCf$\sqrt{t}$ | 88.39 | 9.53 | 14.06 | 21.00 | 31.85 | 22.27 | 31.98 | 16.70 |
| DLMC$\sqrt{t}$ | 85.28 | 12.05 | 17.65 | 24.03 | 36.34 | 24.51 | 36.27 | 16.45 |
| GWG$\frac{t}{t+1}$ | 81.15 | 15.47 | 22.70 | **25.62** | 38.91 | 25.62 | 38.58 | 16.68 |
| DMALA$\frac{t}{t+1}$ | 80.21 | **16.36** | 23.71 | 25.60 | **39.39** | 26.75 | **39.72** | 16.53 |
| PAS$\frac{t}{t+1}$ | 81.02 | 15.62 | 22.65 | 25.59 | 39.28 | 26.08 | 39.48 | 16.69 |
| DLMCf$\frac{t}{t+1}$ | **80.12** | 16.25 | **23.76** | 25.41 | 39.31 | **26.86** | 39.57 | 16.73 |
| DLMC$\frac{t}{t+1}$ | 84.55 | 12.62 | 18.47 | 24.27 | 37.28 | 24.94 | 37.14 | 16.69 |

# 5 Conclusion

*DISCS* is a tailored benchmark for discrete sampling. It implements a range of discrete sampling tasks and state-of-the-art discrete samplers and enables a fair comparison. From the results, we know that DLMC leads in sampling from classical graphical models, PAS leads in solving combinatorial optimization problems, DLMCf and DMALA have the best performance on language models. We believe more efficient discrete samplers can be obtained by designing better discretization of DLD (Sun et al., 2022a). *DISCS* is a convenient tool during this process. The researcher can freely set the configurations for tasks and samplers and *DISCS* will automatically compile the program and run the processes in parallel. Besides, we observe that the choice of the locally balanced weight function should depend on the critical temperature of the target distribution. We believe this observation is insightful and will lead to a deeper understanding of locally balanced samplers.

Of course, *DISCS* does not include all existing tasks or samplers in discrete sampling, for example, the zero order (Xiang et al., 2023) and second order (Sun et al., 2023a) approximation methods. We will keep iterating *DISCS* and more features will be added in the future. We wrap *DISCS* to a JAX library. Researchers can conveniently implement customer tasks or samplers to accelerate their study and, in the meanwhile, contribute the code to *DISCS* for further improvement. We believe *DISCS* will be a powerful tool for researchers and facilitate future research in discrete sampling.

### Acknowledgments

The authors would like to thank Bo Dai, Bethany Wang, Emily Xue for providing helpful discussions, and Pengcheng Ying, Sherry Yang and anonymous reviewers for the helpful comments on the paper.

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

# A Experiments Details

## A.1 Classical Graphical Models

For all the experiments of classical graphical models, we run 100 chains. The chains run in parallel on 4 V100 GPUs, with each GPU managing a mini-batch of 25 chains. We report the performance of all the samplers on Bernoulli Model 7, Categorical Model 8, Ising Model 9, Potts Model 10 and FHMM 11. We study the effect of number of discrete variables (sample dimension), the number of categories for each discrete variable, weight function for locally balanced samplers, and the smoothness/sharpness of different models. Note that the result for BG-2 on Potts 10 and Categorical 8 model with 256 categories are omitted as it takes over 100 hours. The chain length is set as 1 million steps when studying the effect of the number of categories and sample dimension. In the other cases, we use 100k steps as the chain length. For each experiment, as we sample the chains, each sample from each chain at each step is mapped to a lower dimension of 1. The samples are mapped on the same arbitrary sample which we randomly initialize in the beginning of the experiment. We calculate ESS on the mapped samples after the burn-in phase i.e. after the generation of half of the chain. We explain more details on ESS computation in section A.6.1. The average and standard deviation of ESS is computed over all the chains and is reported over the running time and number of energy evaluation of each sampler.

In the following sections, we provide the energy function we used for each of the classical graphical models.

### A.1.1 Factorized Models

Factorized models are the simplest distributions in a discrete space, where each site is independent from the others. Consider the category set of one hot vectors $\mathcal{C} = \{e_1, ..., e_C\}$ and the state space $\mathcal{X} = \mathcal{C}^N$. We have $|\mathcal{C}| = C$ is the number of category and $N$ is the number of variables. The energy function of a factorized model is:

$$f(x) = \sum_{n=1}^{N} \langle x_n, \theta^n \rangle \tag{9}$$

where $\theta^d \in \mathbb{R}^C$. We denote the target distribution as Bernoulli model when $C = 2$ and Categorical model when $C > 2$. We report the results on Bernoulli models and Categorical models in Figure 7 and 8, respectively.

### A.1.2 Ising Models

The Ising model (Ising, 1924) is a mathematical model of ferromagnetism in statistical mechanics. It consists of binary random variables arranged in a lattice graph $G = (V, E)$ and allows a node to interact with its neighbors. The Potts model (Potts, 1952) is a generalization of the Ising model where the random variables are categorical. The energy function for Ising model and Potts model can be described as:

$$f(x) = -\sum_{n=1}^{N} \langle x_n, \theta_n \rangle - \sum_{(i,j) \in E} J_{ij}(x_i, x_j) \tag{10}$$

where we set $\theta^d \in \mathbb{R}^n$, and $J_{ij}(x_i, y_j) = 1_{\{x_i = y_j\}}$. For Ising model, we use $\theta^n \sim \text{Uniform}(-2, 1)$ for the outer part of the lattice graph, and $\theta^n \sim \text{Uniform}(-1, 2)$ for the inner part of the lattice graph. We report the results on Ising model and Potts model in Figure 9, 10.

### A.1.3 Factorial Hidden Markov Model (FHMM)

FHMM (Ghahramani & Jordan, 1995) uses latent variables to characterize time series data. In particular, it assumes the continuous data $y \in \mathbb{R}^L$ is generated by hidden state $x \in \mathcal{C}^{L \times K}$. The probability function is:

$$p(x) = p(x_1) \prod_{l=2}^{L} p(x^t | x^{t-1}), \quad p(y|x) = \prod_{l=1}^{L} \mathcal{N}(y_t; \sum_{k=1}^{K} \langle W_k, x_{l,k} \rangle + b; \sigma^2) \tag{11}$$

In particular, for binary model, we consider $\mathbb{P}(x_1 = 0) = 0.9, \mathbb{P}(x^t = x^{t-1}|x^{t-1}) = 0.8, \sigma = 2.0$. We use $L = 200, K = 50$ for high temperature setting and $L = 1000, K = 10$ in low temperature setting. For categorical model, we use $p(x_1|x_1 \neq 0)$ and $p(x^t|x^{t-1}, x^t \neq x^{t-1})$ as uniform distribution and we use $L = 200, K = 10$ with category number $C = 4, 8$. We report the results in Figure 11.

## A.2 Combinatorial Optimization

Table 3: Synthetic data statistics.

| Name | MIS | | Max Clique | Maxcut | |
|---|---|---|---|---|---|
| | ER-[700-800] | ER-[9000-11000] | RB | ER | BA |
| Max # nodes | 800 | 10,915 | 475 | 1,100 | 1,100 |
| Max # edges | 47,885 | 1,190,799 | 90,585 | 91,239 | 4,384 |
| # Test instances | 128 | 16 | 500 | 1,000 | 1,000 |

Table 4: Real-world data statistics.

| Name | MIS | Max Clique | Maxcut | Balanced Graph Partition | | | | |
|---|---|---|---|---|---|---|---|---|
| | SATLIB | Twitter | Optsicom | MNIST | VGG | ALEXNET | RESNET | INCEPTION |
| Max # nodes | 1,347 | 247 | 125 | 414 | 1,325 | 798 | 20,586 | 27,114 |
| Max # edges | 5,978 | 12,174 | 375 | 623 | 2,036 | 1,198 | 32,298 | 40,875 |
| # Test instances | 500 | 196 | 10 | 1 | 1 | 1 | 1 | 1 |

Here we first provide the experimental details for the combinatorial optimization problems, Maximum Independent Set (MIS), Max Clique, MaxCut and, Balanced Graph Partition. The statistics of the synthetic datasets, including the maximum number of nodes/edges in a graph, and the number of test instances are reported in 3. Additionally the statistics of real-world graphs are in 4. For MaxCut-ba, all Balanced Graph Partition and MIS graphs, we use 32 as the number of chains and for MaxCut-optsicom, MaxCut-er, and all Max Clique graphs we use 16. The data used for these experiments could be found at DISCS DATA.

We run all the experiments on 8 V100 GPUs in parallel. For only MaxCut Optsicom graph, we use 2 V100 GPUs. The test instances are divided evenly between the GPUs and are run in parallel. For each experiment, we report the average and standard deviation of the best solution found over the number of test instances along with the end-to-end run time (in seconds) of each in tables. We report the results for all the samplers and plot their solution as the chain is being generated over M-H step and the running time. Note that for reporting the standard deviation for the plots, for better visualization, we clip the found solutions by a minimum of zero (removing the penalty effect of the optimization objective). However, the tables include the accurate standard deviation of the best solution found over the instances.

In the following sections, we provide the energy function we use for each of the combinatorial optimization problems. For a graph $G = (V, E)$ we label the nodes in $V$ from 1 to $d$. The adjacency matrix is represented as $A$. For a weighted graph, we simply let $A_{ij}$ denote the edge weight between node $i$ and $j$. For constraint problems, we follow Sun et al. (2022c) to select penalty coefficient $\lambda$ as the minimum value of $\lambda$ such that $x^* := \arg\min f(x)$ is achieved at $x^*$ satisfying the original constraints. Such a choice of the coefficient guarantees the target distribution converges to the optimal solution of the original CO problems while keeping the target distribution as smooth as possible.

### A.2.1 Maximum Independent Set (MIS)

The MIS has the integer programming formulation as

$$\min_{x \in \{0,1\}^d} -\sum_{i=1}^{d} c_i x_i, \quad \text{s.t. } x_i x_j = 0, \ \forall (i, j) \in E \tag{12}$$

We use the corresponding energy function in the following quadratic form:

$$f(x) := -c^T x + \lambda \frac{x^T A x}{2} \tag{13}$$

In our experiments $c$ equals to 1 and we use $\lambda = 1.0001$. In post processing, we iteratively go through all nodes $x_i$ for $i = 1, ..., d$. If there exists $x_j = 1$ for $(x_i, x_j) \in E$, we flip its value $x_j = 0$. After post processing, the state $x$ is guaranteed to be feasible in the original MIS problem. We provide the average and standard deviation of the best solutions over all the number of instances along with their corresponding running time (s) at 5. The plots of the experiments could be found at 12.

Table 5: MIS.

| Sampler | Graphs Density | ER[700-800] | | | | | ER[9000-11000] | SATLIB |
|---------|---------|---------|---------|---------|---------|---------|---------|---------|
| | | 0.05 | 0.10 | 0.15 | 0.20 | 0.25 | 0.15 | 0.15 |
| HB-10-1 | Size | $100.374 \pm 2.073$ | $58.750 \pm 1.172$ | $41.812 \pm 0.864$ | $32.344 \pm 0.643$ | $26.469 \pm 0.749$ | $277.149 \pm 3.206$ | $422.427 \pm 14.403$ |
| | Time(s) | 426.185 | 390.810 | 684.590 | 414.067 | 429.879 | 15139.425 | 5381.857 |
| BG-2 | Size | $102.468 \pm 2.015$ | $60.000 \pm 0.707$ | $42.820 \pm 0.765$ | $32.250 \pm 0.500$ | $27.312 \pm 0.634$ | $316.170 \pm 3.187$ | $422.200 \pm 14.390$ |
| | Time(s) | 291.427 | 290.042 | 562.986 | 295.024 | 288.109 | 13079.125 | 3027.204 |
| RWM | Size | $97.186 \pm 1.943$ | $56.249 \pm 0.968$ | $40.429 \pm 0.777$ | $31.219 \pm 0.599$ | $25.594 \pm 0.605$ | $-555.674 \pm 359.008$ | $420.284 \pm 14.263$ |
| | Time(s) | 284.092 | 293.517 | 499.577 | 297.140 | 281.772 | 12401.737 | 2955.729 |
| GWG-nA | Size | $104.812 \pm 1.590$ | $62.125 \pm 0.739$ | $44.383 \pm 0.830$ | $34.812 \pm 0.583$ | $28.187 \pm 0.463$ | $367.310 \pm 4.383$ | $422.971 \pm 14.407$ |
| | Time(s) | 278.885 | 308.873 | 737.671 | 303.435 | 310.551 | 24698.296 | 3540.670 |
| DMALA | Size | $104.750 \pm 1.803$ | $62.031 \pm 0.684$ | $44.195 \pm 0.781$ | $34.375 \pm 0.599$ | $28.031 \pm 0.467$ | $357.058 \pm 9.622$ | $423.641 \pm 14.506$ |
| | Time(s) | 291.271 | 292.131 | 714.614 | 297.848 | 298.732 | 24769.380 | 3465.343 |
| PAS | Size | $105.062 \pm 1.560$ | $62.250 \pm 0.790$ | $44.570 \pm 0.669$ | $34.719 \pm 0.6242$ | $28.500 \pm 0.500$ | $377.123 \pm 4.498$ | $424.143 \pm 14.531$ |
| | Time(s) | 299.004 | 310.765 | 759.372 | 299.569 | 308.475 | 25242.166 | 4826.039 |
| DLMCF | Size | $104.450 \pm 1.561$ | $62.219 \pm 0.926$ | $44.078 \pm 0.746$ | $34.469 \pm 0.558$ | $28.125 \pm 0.414$ | $354.121 \pm 10.683$ | $423.387 \pm 14.441$ |
| | Time(s) | 291.366 | 301.554 | 726.287 | 302.667 | 300.413 | 24892.216 | 3679.425 |
| DLMC | Size | $104.844 \pm 1.769$ | $62.187 \pm 0.882$ | $44.273 \pm 0.788$ | $34.500 \pm 0.707$ | $28.281 \pm 0.450$ | $355.058 \pm 10.128$ | $423.479 \pm 14.483$ |
| | Time(s) | 293.235 | 294.975 | 725.326 | 294.688 | 299.884 | 24976.312 | 3523.320 |

We also conduct experiments to justify the results are robust regarding the choice of the penalty coefficient. In Figure 13, we use penalty coefficient $\lambda \in \{1.001, 1.01, 1.1, 2\}$ on ER-[700-800] graphs with density $\{0.05, 0.10, 0.15, 0.20, 0.25\}$. We also use a dashed line to represent the optimal value obtained by running Gurobi-10 for 1 hour. From the results, we can observe that 1) PAS consistently obtains the best results, 2) locally balanced samplers have results consistently better than traditional sampler and Gurobi.

### A.2.2 Max Clique

The max clique problem is equivalent to MIS on the dual graph. In our experiments $c$ equals to 1.

$$\min_{x \in \{0,1\}^d} - \sum_{i=1}^{d} c_i x_i, \quad \text{s.t. } x_i x_j = 0, \ \forall (i, j) \notin E \tag{14}$$

The energy function is

$$f(x) := -c^T x + \frac{\lambda}{2} \left( \mathbf{1}^\top x \cdot (\mathbf{1}^\top x - 1) - x^T A x \right) \tag{15}$$

In our experiments $c$ equals to 1 and we use $\lambda = 1.0001$. In post processing, we iteratively go through all nodes $x_i$ for $i = 1, ..., d$. If there exists $x_j = 1$ for $(x_i, x_j) \notin E$, we flip its value $x_j = 0$. After post processing, the state $x$ is guaranteed to be feasible in the original Max Clique problem. We provide the average and the standard deviation of the best solutions over all number of instances along with their corresponding running time at 6. The plots of the experiments could be found at 14.

### A.2.3 MaxCut

We optimize the following problem:

$$\min_{x \in \{-1,1\}^d} - \sum_{(i,j) \in E} A_{i,j} \left( \frac{1 - x_i x_j}{2} \right) \tag{16}$$

Note that for simplicity each dimension of $x$ is selected from $\{-1, 1\}$. To represent the corresponding energy function for $x \in \{0, 1\}^d$, we have

$$f(x) := - \sum_{(i,j) \in E} A_{i,j} \left( \frac{1 - (2x_i - 1)(2x_j - 1)}{2} \right) \tag{17}$$

In our experiments $A_{ij}$ equals to 1. Since the problem is always feasible, the post processing is an identity map. We provide the average and standard deviation of the best solutions over all number of instances along with their corresponding running time at 7. The plots of the experiments could be found at 15.

Table 6: Max Clique.

| Sampler | Results | RB | TWITTER |
|---|---|---|---|
| HB-10-1 | Ratio $\alpha$ | $0.850 \pm 0.0620$ | $0.966 \pm 0.056$ |
| | Time(s) | 1724.893 | 6.817 |
| BG-2 | Ratio $\alpha$ | $0.859 \pm 0.061$ | $0.995 \pm 0.030$ |
| | Time(s) | 1592.808 | 6.327 |
| RWM | Ratio $\alpha$ | $0.841 \pm 0.0633$ | $0.584 \pm 0.484$ |
| | Time(s) | 1683.397 | 5.664 |
| GWG-nA | Ratio $\alpha$ | $0.878 \pm 0.062$ | $0.999 \pm 0.010$ |
| | Time(s) | 2525.801 | 6.032 |
| DMALA | Ratio $\alpha$ | $0.876 \pm 0.0620$ | $0.999 \pm 0.004$ |
| | Time(s) | 2561.617 | 6.190 |
| PAS | Ratio $\alpha$ | $0.878 \pm 0.0618$ | $0.999 \pm 0.011$ |
| | Time(s) | 2542.538 | 6.160 |
| DLMCF | Ratio $\alpha$ | $0.871 \pm 0.061$ | $0.999 \pm 0.011$ |
| | Time(s) | 2532.835 | 5.988 |
| DLMC | Ratio $\alpha$ | $0.875 \pm 0.062$ | $0.999 \pm 0.009$ |
| | Time(s) | 2639.588 | 6.124 |

Table 7: Maxcut.

| Sampler | Results | BA | | | | | | | ER | | | OPTSICOM |
|---|---|---|---|---|---|---|---|---|---|---|---|---|
| | | 16-20 | 32-10 | 64-75 | 128-150 | 256-300 | 512-600 | 1024-1100 | 256-300 | 512-600 | 1024-1100 | |
| HB-10-1 | Ratio $\alpha$ | 1.000±0.000 | 1.000±0.000 | 1.000±0.000 | 1.000±0.000 | 1.000±0.001 | 1.008±0.005 | 1.014±0.004 | 1.020±0.017 | 1.000±0.001 | 0.998±0.001 | 1.000±0.000 |
| | Time(s) | 742.568 | 754.613 | 749.626 | 783.278 | 792.338 | 1143.302 | 1890.534 | 331.019 | 416.002 | 1488.382 | 75.347 |
| BG-2 | Ratio $\alpha$ | 1.000±0.000 | 1.000±0.000 | 1.000±0.000 | 1.000±0.000 | 1.000±0.000 | 1.009±0.005 | 1.014±0.004 | 1.021±0.018 | 1.001±0.001 | 0.999±0.001 | 1.000±0.000 |
| | Time(s) | 517.183 | 538.258 | 550.082 | 553.863 | 531.720 | 578.991 | 1157.571 | 269.116 | 337.014 | 1295.219 | 17.050 |
| RWM | Ratio $\alpha$ | 0.998±0.000 | 1.000±0.000 | 1.000±0.000 | 1.000±0.000 | 0.999±0.001 | 1.005±0.005 | 1.007±0.004 | 1.019±0.017 | 0.997±0.002 | 0.996±0.001 | 1.000±0.000 |
| | Time(s) | 534.215 | 534.615 | 528.641 | 558.608 | 541.302 | 574.778 | 1065.852 | 267.071 | 333.402 | 1266.630 | 58.960 |
| GWG-nA | Ratio $\alpha$ | 1.000±0.000 | 1.000±0.000 | 1.000±0.000 | 1.000±0.000 | 1.000±0.000 | 1.010±0.005 | 1.017±0.004 | 1.021±0.017 | 1.002±0.001 | 1.001±0.001 | 1.000±0.000 |
| | Time(s) | 522.094 | 531.425 | 578.917 | 551.923 | 545.634 | 724.721 | 1427.577 | 264.202 | 466.199 | 1666.021 | 80.124 |
| DMALA | Ratio $\alpha$ | 1.000±0.000 | 1.000±0.000 | 1.000±0.000 | 1.000±0.000 | 1.000±0.000 | 1.010±0.005 | 1.018±0.004 | 1.021±0.017 | 1.002±0.001 | 1.002±0.001 | 1.000±0.000 |
| | Time(s) | 531.433 | 538.938 | 568.224 | 549.026 | 544.568 | 750.909 | 1490.872 | 277.855 | 461.179 | 1643.135 | 53.509 |
| PAS | Ratio $\alpha$ | 1.000±0.000 | 1.000±0.000 | 1.000±0.000 | 1.000±0.000 | 1.000±0.000 | 1.010±0.005 | 1.018±0.004 | 1.021±0.017 | 1.002±0.001 | 1.002±0.001 | 1.000±0.000 |
| | Time(s) | 519.842 | 538.814 | 550.035 | 550.578 | 580.051 | 940.408 | 1917.954 | 278.005 | 543.607 | 1689.071 | 59.213 |
| DLMCF | Ratio $\alpha$ | 1.000±0.000 | 1.000±0.000 | 1.000±0.000 | 1.000±0.000 | 1.000±0.000 | 1.010±0.005 | 1.018±0.004 | 1.021±0.017 | 1.002±0.001 | 1.001±0.005 | 1.000±0.000 |
| | Time(s) | 521.592 | 526.289 | 545.877 | 557.564 | 533.119 | 765.719 | 1510.380 | 272.841 | 452.252 | 1639.539 | 52.552 |
| DLMC | Ratio $\alpha$ | 1.000±0.000 | 1.000±0.000 | 1.000±0.000 | 1.000±0.000 | 1.000±0.000 | 1.010±0.005 | 1.018±0.004 | 1.021±0.017 | 1.002±0.001 | 1.002±0.001 | 1.000±0.000 |
| | Time(s) | 531.003 | 550.118 | 543.287 | 544.611 | 542.677 | 765.104 | 1564.198 | 271.262 | 451.080 | 1642.223 | 53.368 |

### A.2.4 Balanced Graph Partition

We find the following objective for balanced graph partition gives the best result:

$$f(x) := \sum_{s=1}^{k} \sum_{(i,j)\in E} \mathbb{I}\left(x_i \neq x_j \&\&(x_i = s || x_j = s)\right) + \sum_{s=1}^{k} \left(d/k - \sum_{i=1}^{d} \mathbb{I}(x_i = s)\right)^2 \quad (18)$$

where $k$ is the number of partitions. Since the problem is always feasible, the post processing is identity map. We provide the edge cut ratio and balanceness of the best samples over all the chains at 8. Further details on the calculated metrics could be found at A.6.2.

### A.3 Energy Based Generative Models

### A.3.1 Restricted Boltzmann Machine

The RBM is an unnormalized latent variable model, with a visible random variable $v \in \mathcal{C}^N$ and a hidden random variable $h \in \{0,1\}^M$. When $v$ is binary, we call it a binary RBM (binRBM) and when $v$ is categorical, we call it a categorical RBM (catRBM). The energy function of both binRBM and catRBM (Tran et al., 2011) can be written as:

$$f(v) = \sum_{h} \left[ -\sum_{n=1}^{N} \langle v_n, \theta_n \rangle - \sum_{m=1}^{M} \beta_m h_m - \sum_{d,m} \langle h_m \theta_{m,d}, v_n \rangle \right] \quad (19)$$

Unlike the previous three models, where the parameters are hand designed, we train binary RBM on MNIST (LeCun, 1998) and categorical RBM on Fashion-MNIST (Xiao et al., 2017a) using contrastive divergence Hinton (2002). Across all settings, we have $D = 784$. For binary models, we use $M = 25$ for high temperature setting and $M = 200$ for low temperature setting. For categorical models, we use $M = 100$. We report the results in Figure 16. The experimental setup is similar to classical graphical models.

Table 8: Balanced graph partition.

| Metric | Samplers | VGG | MNIST-conv | ResNet | AlexNet | Inception-v3 |
|---|---|---|---|---|---|---|
| | HB-10-1 | 0.050 | 0.046 | 0.050 | 0.037 | 0.065 |
| | BG-2 | **0.048** | **0.045** | 0.050 | 0.038 | 0.069 |
| | RWM | 0.054 | 0.046 | 0.092 | 0.052 | 0.117 |
| | GWG | 0.102 | 0.046 | 0.159 | 0.063 | 0.164 |
| | DMALA | 0.084 | 0.058 | 0.178 | 0.063 | 0.176 |
| Edge cut ratio ↓ | DMALA-nA | 0.059 | **0.045** | 0.048 | 0.039 | 0.054 |
| | PAS | 0.053 | **0.045** | **0.047** | **0.037** | **0.052** |
| | PAS-nA | 0.084 | 0.050 | 0.138 | 0.053 | 0.144 |
| | DLMCF | 0.086 | 0.063 | 0.178 | 0.053 | 0.176 |
| | DLMCF-nA | 0.092 | 0.069 | 0.048 | 0.085 | **0.052** |
| | DLMC | 0.105 | 0.056 | 0.183 | 0.097 | 0.182 |
| | DLMC-nA | 0.113 | 0.048 | 0.082 | 0.091 | 0.086 |
| | HB-10-1 | 0.999 | 0.999 | 0.999 | 0.999 | 0.999 |
| | BG-2 | 0.999 | 0.997 | 0.999 | 0.999 | 0.999 |
| | RWM | 0.999 | 0.998 | 0.999 | 0.999 | 0.999 |
| | GWG | 0.999 | 0.997 | 0.999 | 0.999 | 0.999 |
| | DMALA | 0.999 | 0.998 | 0.999 | 0.999 | 0.999 |
| Balanceness ↑ | DMALA-nA | 0.999 | 0.997 | 0.999 | 0.999 | 0.999 |
| | PAS | 0.999 | 0.997 | 0.999 | 1.000 | 0.999 |
| | PAS-nA | 0.999 | 0.998 | 0.999 | 0.999 | 0.999 |
| | DLMCF | 0.999 | 0.997 | 0.999 | 0.999 | 0.999 |
| | DLMCF-nA | 0.999 | 0.995 | 0.999 | 0.999 | 0.999 |
| | DLMC | 0.999 | 0.994 | 0.999 | 0.999 | 0.999 |
| | DLMC-nA | 0.999 | 0.993 | 0.999 | 0.999 | 0.999 |

### A.3.2 Deep residual network

In this experiment, we train a deep residual network on MNIST, Omniglot and Caltech dataset. The models checkpoints could be found at DISCS DATA. We use all the samplers to sample from the trained energy based generative models. We use the chain length of 10k and number of chains of 100. We randomly selected one chain from the 100 chains and save its sample after each 1k steps, giving us 10 images per each chain for each sampler 17. We can see that locally balanced samplers are able to generate higher quality images faster and visit more diverse modalities.

### A.3.3 Text Infilling

The experimental set up is explained in detail at 4.4. Here we additionally provide the performance of the locally balanced samplers in their non adaptive condition observed at 9. The data used for this experiment could be found at DISCS DATA.

### A.4 *DISCS* Code

The source code is open source at DISCS with extensive documentation.

### A.5 Data-Set

The data used in this paper is available at DISCS DATA. Under the data set folder, you can find:

- Restricted Boltzmann Machine (RBM) checkpoints, more specifically two binary RBM checkpoints trained on MNIST dataset, one with 25 hidden dimensions and other with 200. Two categorical RBM checkpoints, trained on Fashion-MNIST data set with 4 and 8 categories. 16

- Deep residual network checkpoints, more specifically three checkpoints trained on MNIST, Omniglot and Caltech dataset.

- Graphs data used for combinatorial optimization experiments found at sco directory.

- Text infilling data used for text infilling task experimentation.

Table 9: Quantative results on text infilling. The reference text for computing the Corpus BLEU is the combination of WT103 and TBC.

| Methods | Self-BLEU ($\downarrow$) | Unique $n$-grams (%) ($\uparrow$) | | | | | | Corpus BLEU ($\uparrow$) |
| | | Self | | WT103 | | TBC | | |
| | | $n=2$ | $n=3$ | $n=2$ | $n=3$ | $n=2$ | $n=3$ | |
|---|---|---|---|---|---|---|---|---|
| RWM | 92.41 | 6.26 | 9.10 | 18.97 | 26.73 | 19.33 | 26.67 | 16.24 |
| GWG$\sqrt{t}$ | 85.93 | 11.22 | 17.14 | 23.16 | 35.56 | 23.58 | 35.56 | 16.75 |
| GWG$\frac{t}{t+1}$ | 81.15 | 15.47 | 22.70 | 25.62 | 38.91 | 25.62 | 38.58 | 16.68 |
| DMALA-nA$\sqrt{t}$ | 83.99 | 13.26 | 19.52 | 24.33 | 36.40 | 25.30 | 36.40 | 16.37 |
| DMALA-nA$\frac{t}{t+1}$ | 80.44 | 15.86 | 23.58 | 25.79 | 39.88 | 26.57 | 40.20 | 16.64 |
| DMALA$\sqrt{t}$ | 85.88 | 11.58 | 17.14 | 22.07 | 34.08 | 23.22 | 34.15 | 17.06 |
| DMALA$\frac{t}{t+1}$ | 80.21 | 16.36 | 23.71 | 25.60 | 39.39 | 26.75 | 39.72 | 16.53 |
| PAS$\sqrt{t}$ | 85.39 | 11.37 | 17.60 | 22.61 | 35.53 | 23.65 | 35.47 | 16.57 |
| PAS$\frac{t}{t+1}$ | 81.02 | 15.62 | 22.65 | 25.59 | 39.28 | 26.08 | 39.48 | 16.69 |
| DLMCf-nA$\sqrt{t}$ | 91.57 | 7.25 | 10.42 | 19.53 | 28.31 | 20.13 | 28.18 | 16.56 |
| DLMCf-nA$\frac{t}{t+1}$ | 81.66 | 15.31 | 21.78 | 26.39 | 39.56 | 27.60 | 39.69 | 16.31 |
| DLMCf$\sqrt{t}$ | 88.39 | 9.53 | 14.06 | 21.00 | 31.85 | 22.27 | 31.98 | 16.70 |
| DLMCf$\frac{t}{t+1}$ | 80.12 | 16.25 | 23.76 | 25.41 | 39.31 | 26.86 | 39.57 | 16.73 |
| DLMC-nA$\sqrt{t}$ | 83.74 | 12.74 | 19.64 | 24.27 | 37.27 | 24.94 | 37.34 | 16.73 |
| DLMC-nA$\frac{t}{t+1}$ | 82.26 | 14.18 | 21.41 | 25.51 | 39.10 | 26.18 | 39.29 | 16.55 |
| DLMC$\sqrt{t}$ | 85.28 | 12.05 | 17.65 | 24.03 | 36.34 | 24.51 | 36.27 | 16.45 |
| DLMC$\frac{t}{t+1}$ | 84.55 | 12.62 | 18.47 | 24.27 | 37.28 | 24.94 | 37.14 | 16.69 |

## A.6 Metrics and Evaluation Methods

Depending on the task we are solving, we use different metrics and evaluation methods to measure the quality of the generated samples. Below we explain in detail the process of evaluation for different tasks in the benchmark. Additionally, referring to an example provided in 4.4, *DISCS* is structured in a way that researchers can easily plug in new evaluation methods and define their own domain specific scores. For further code implementation details, you can refer to DISCS-Evaluator

### A.6.1 Classical Graphical Models

For each of the experiments of classical graphical models, we run multiple chains in parallel. We rely on Effective Sample Size (ESS) to measure the efficiency of different samplers. We compute the ESS using *tfp.substrates.jax.mcmc.effective_sample_size* from Python *TensorFlow* library on the second half of the chain after the burn-in phase. The average and the standard deviation of the ESS is computed and reported over the chains.

### A.6.2 Combinatorial Optimization

For the combinatorial optimization problems, for each of the test instances, we run multiple chains in parallel. Throughout the chain generation, we keep track of the best solution over all the chains for each of the instances. We report the average and standard deviation of the best solutions found over the test instances for each problem.

For balanced graph partition problem, we additionally store the sample resulting in the best solution of the optimization problem and do further post processing on it. Let $n$ be the number of nodes and the $g$ number of disjoint sets. For each of the instances, we load the corresponding graph and the best solution and compute the edge cut ratio as the ratio of the cut to the total number of edges and the balanceness as one minus the MSE of a number of nodes in every partition and balances partition $n/g$.

### A.6.3 Energy Based Generative Models

For the text infilling task, we evaluate the generated text from two perspectives, diversity, and quality. For diversity, we use self-BLEU (Zhu et al., 2018) and the number of unique n-grams (Wang & Cho, 2019) to measure the difference between the generated sentences. For quality, we measure the BLEU score (Papineni et al., 2002) between the generated texts and the original dataset, which is

the combination of TBC and WikiText-103. We rely on *Natural Language Toolkit (NLTK)* Python library, more specifically *nltk.translate.bleu_score* and *nltk.util.ngrams* to compute the metrics above.

To gain insights into the performance and the quality of the generated samples we also rely on visual representation 17. As the chains are generated, the samples are saved as images at different time intervals. The images could provide insights on the performance of the sampler in terms of the quality and diversity of generated samples.

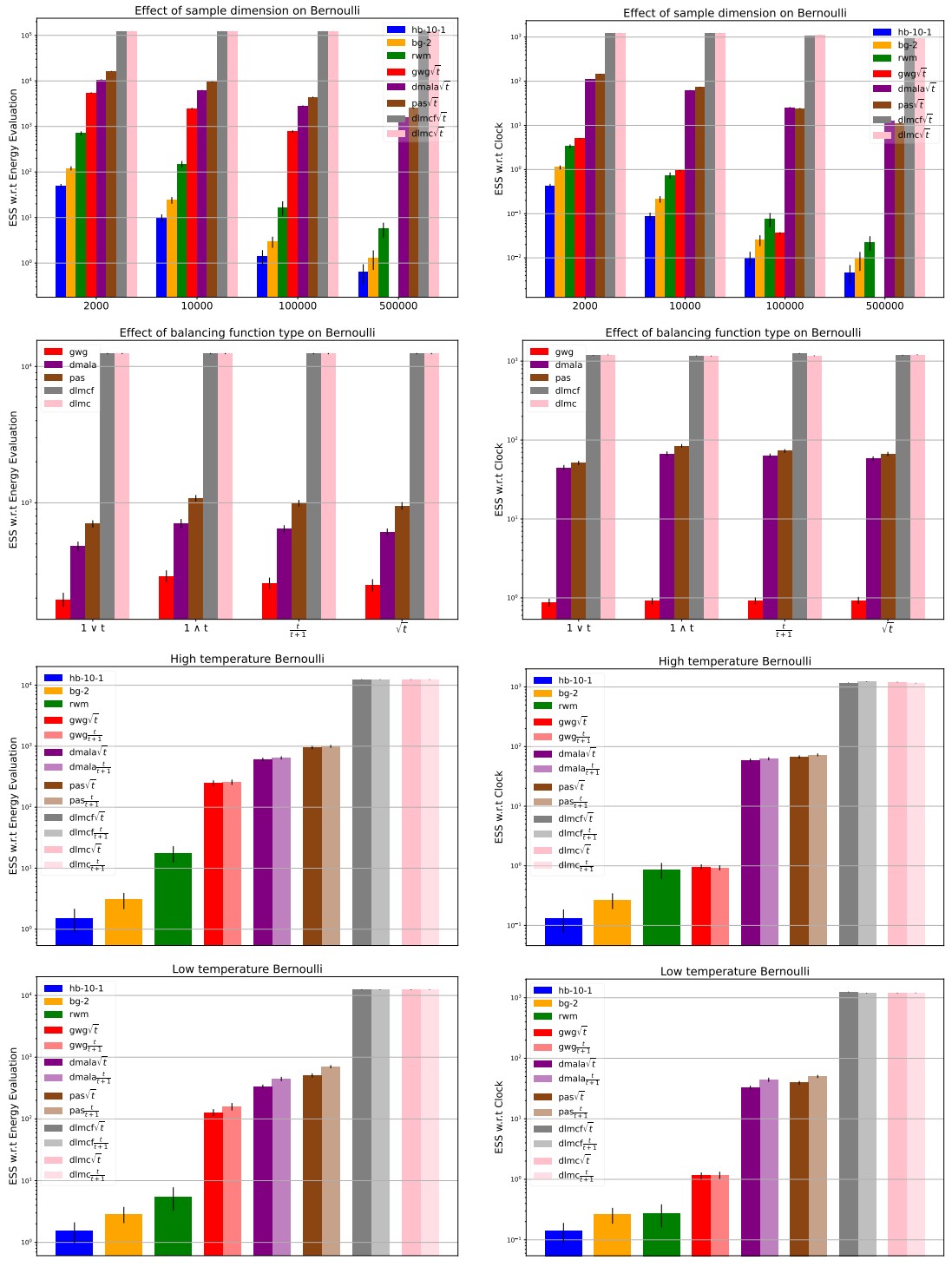

Figure 7: Results of Bernoulli Models

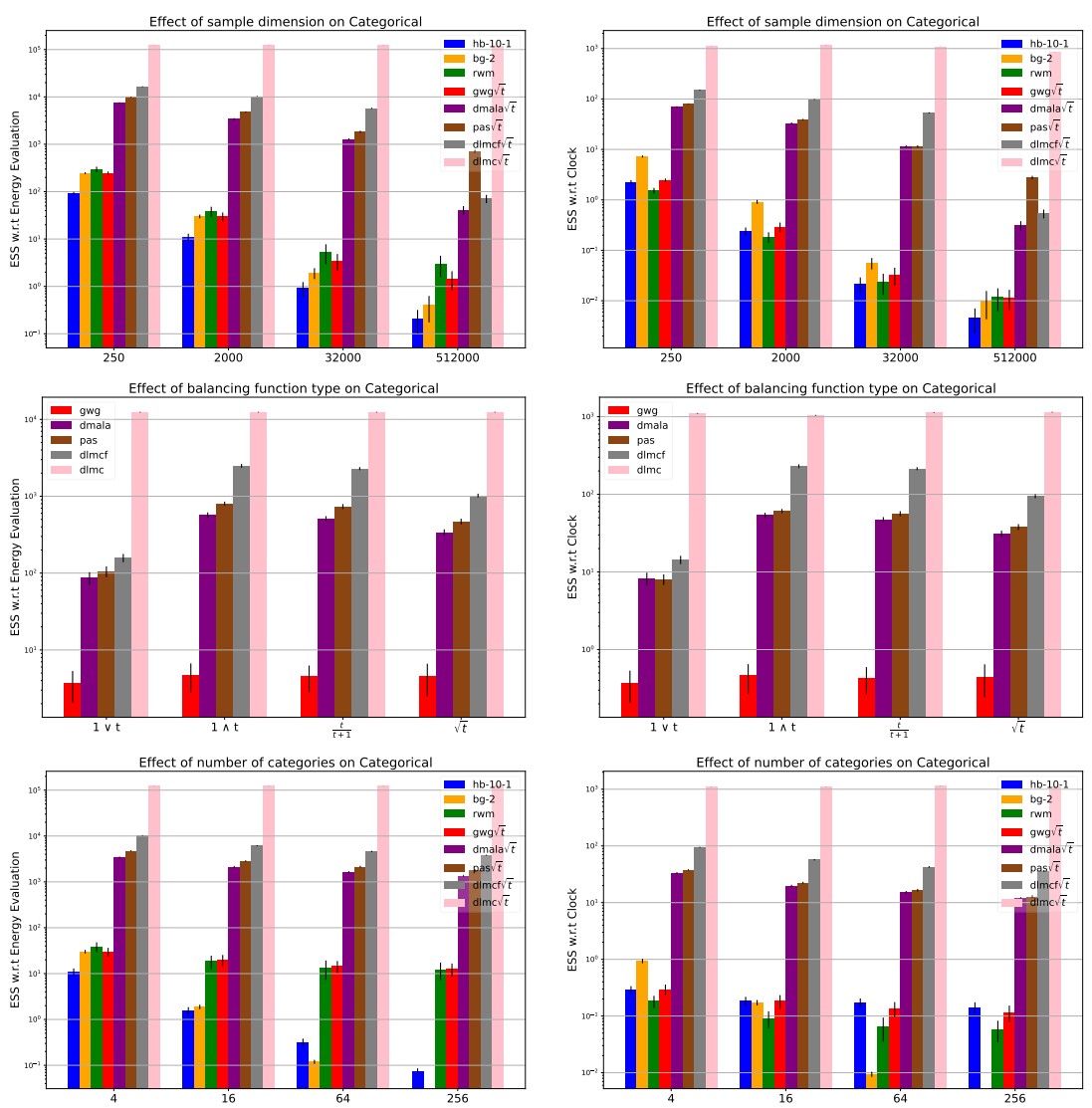

Figure 8: Results of Categorical model

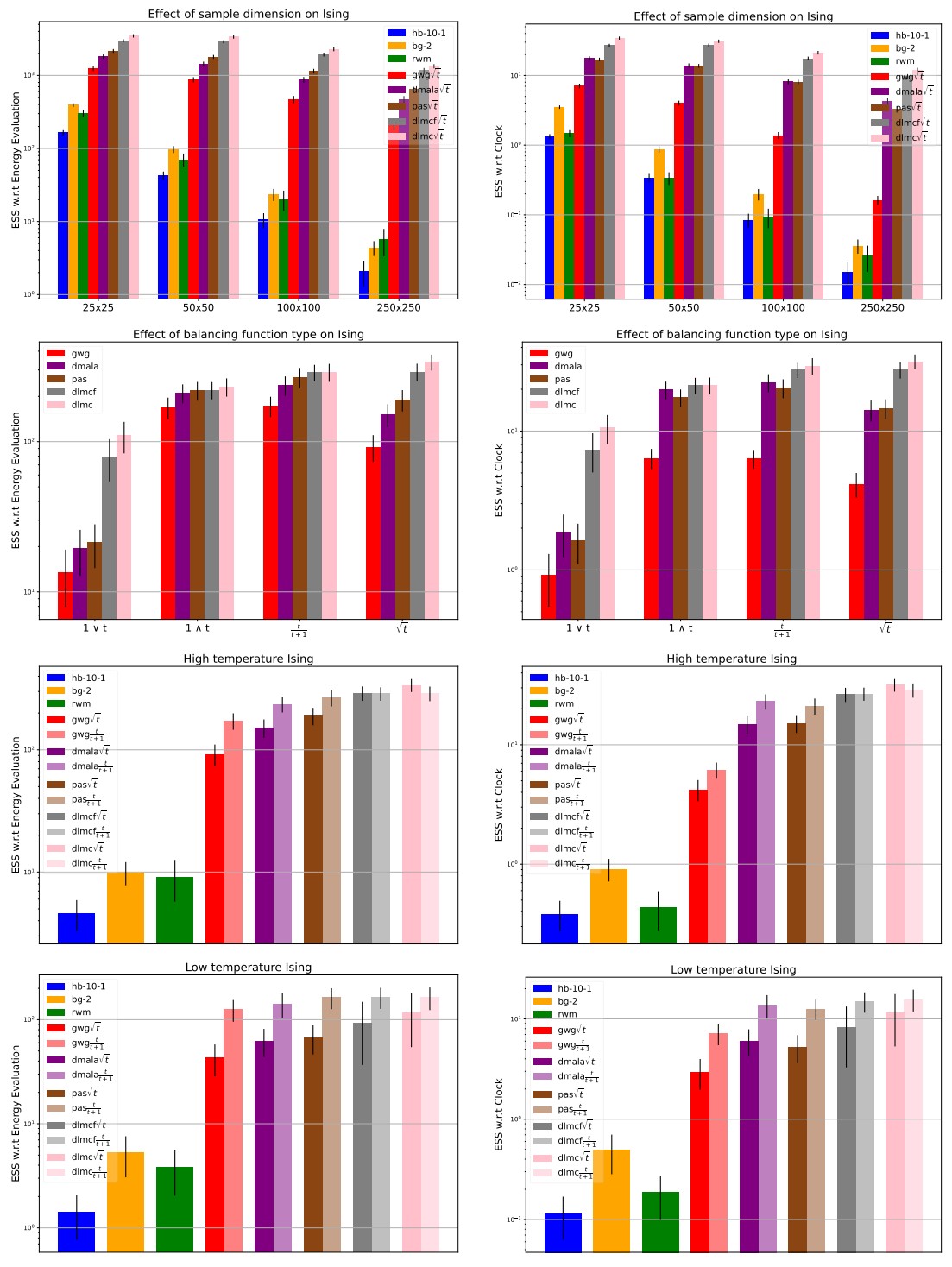

Figure 9: Results of Ising model

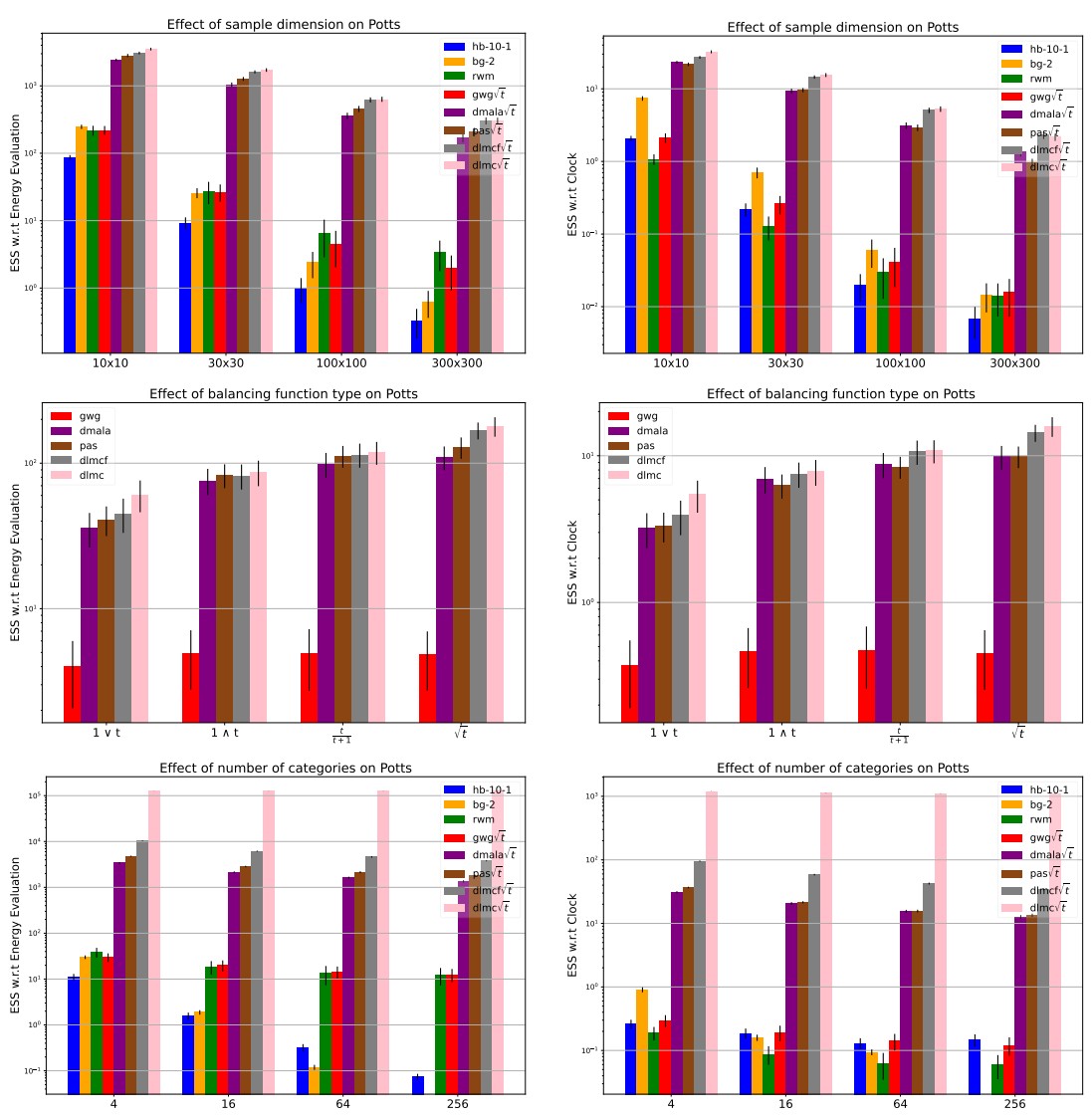

Figure 10: Results of Potts model

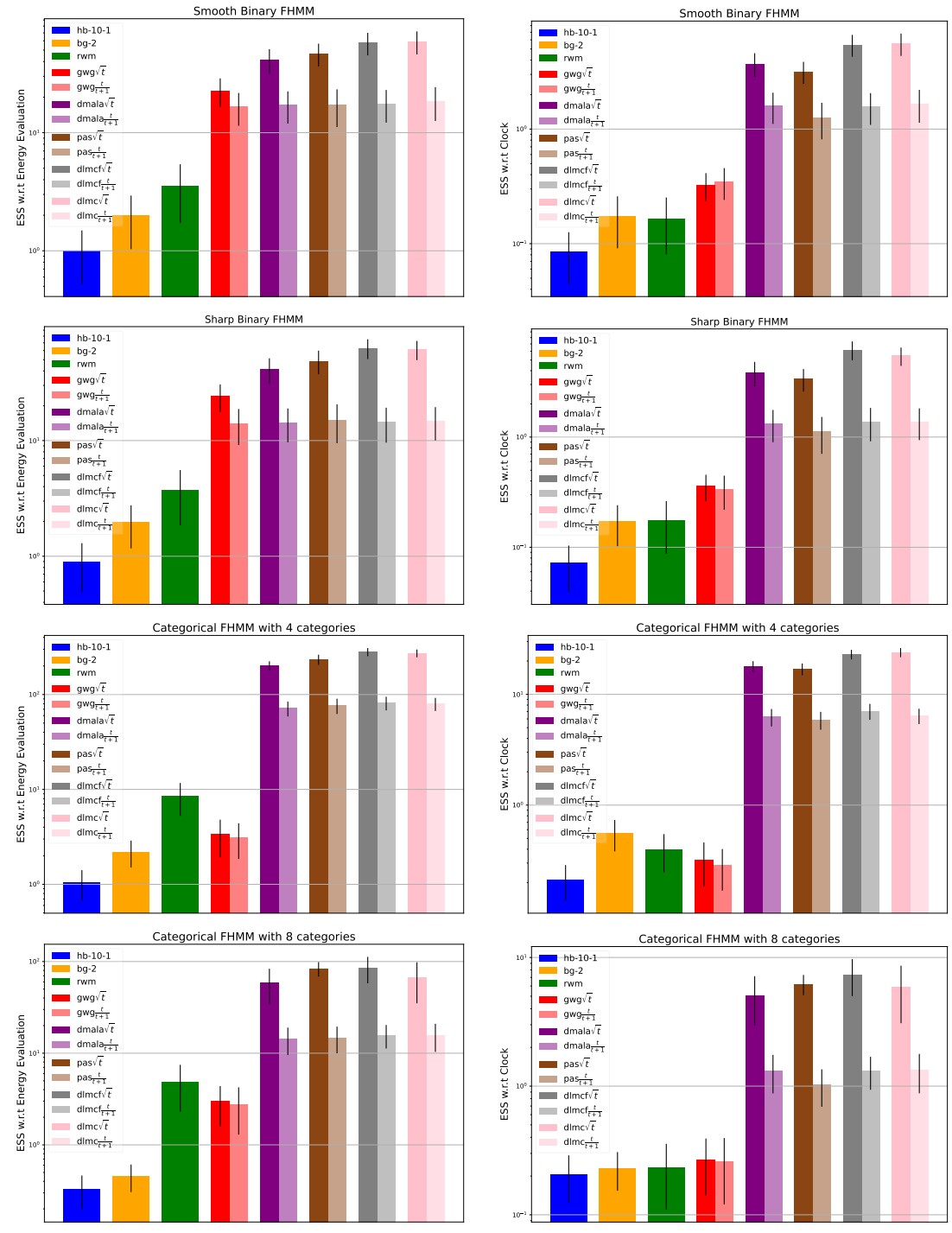

Figure 11: Results of FHMMs

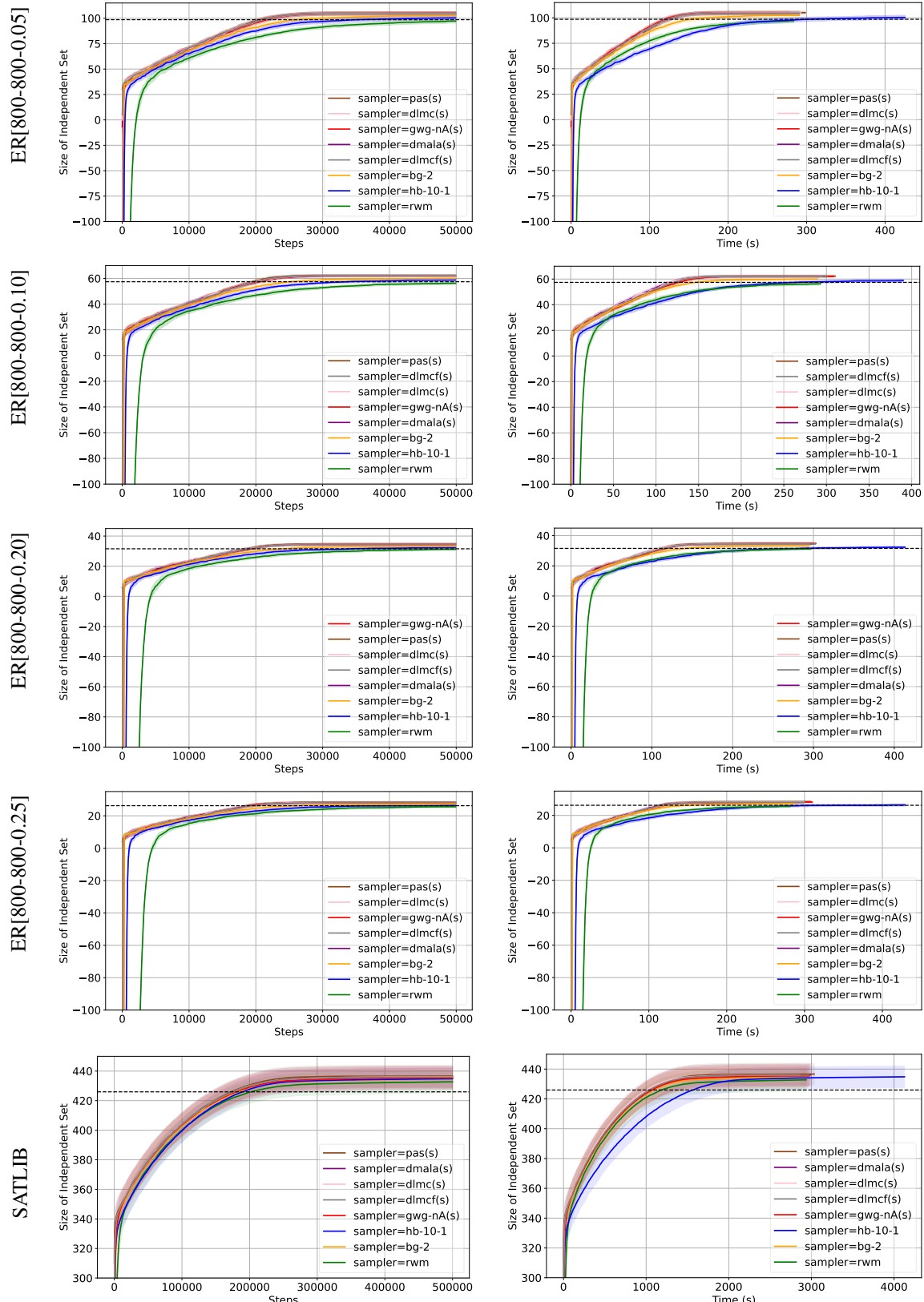

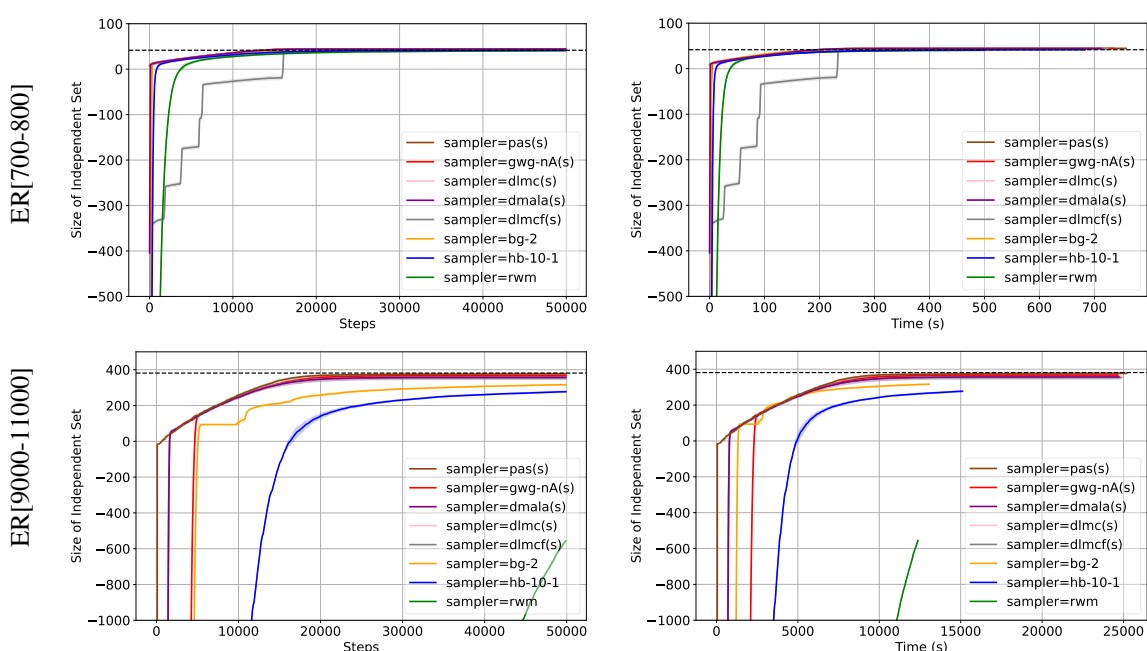

Figure 12: Solving progress on MIS

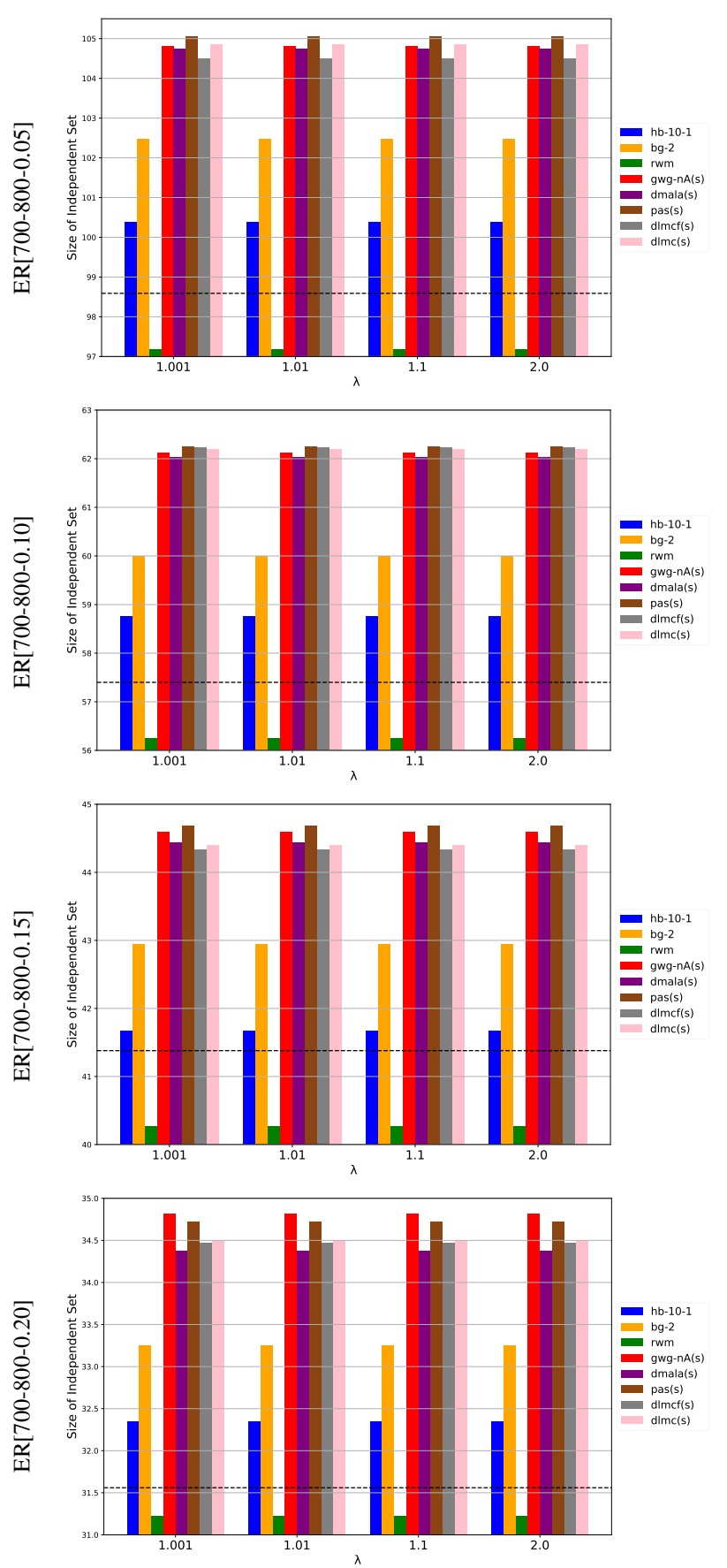

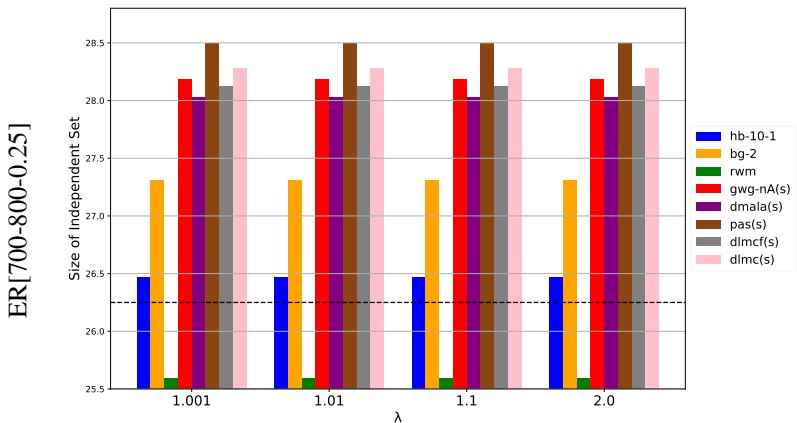

Figure 13: Reuslts on MIS: effect of penalty coefficient. The dashed line represents the best result obtained by running Gurobi for 1 hour.

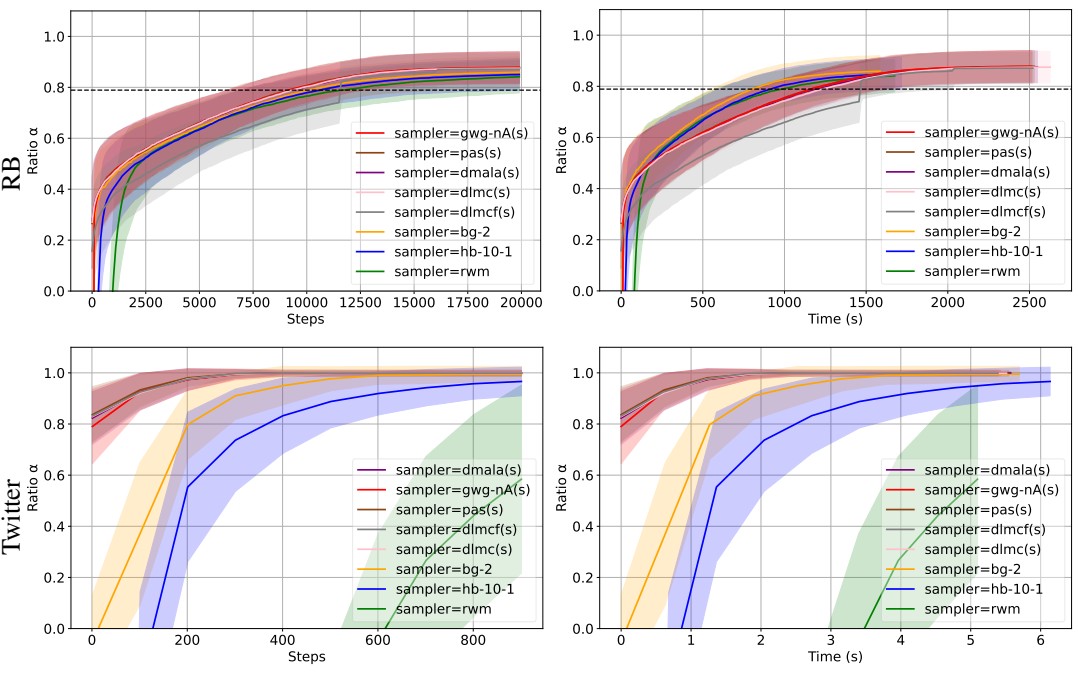

Figure 14: Solving progress on Max Clique

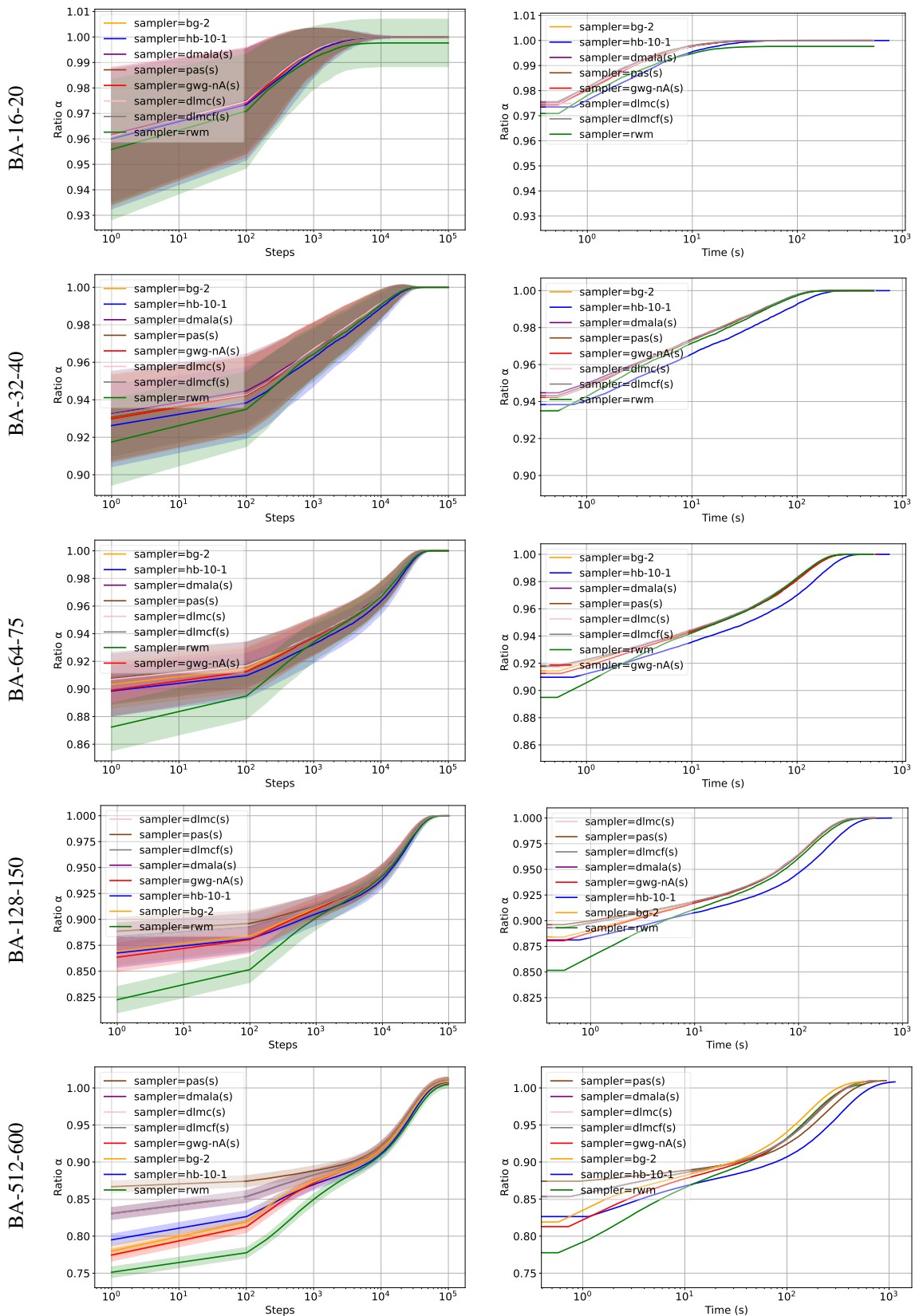

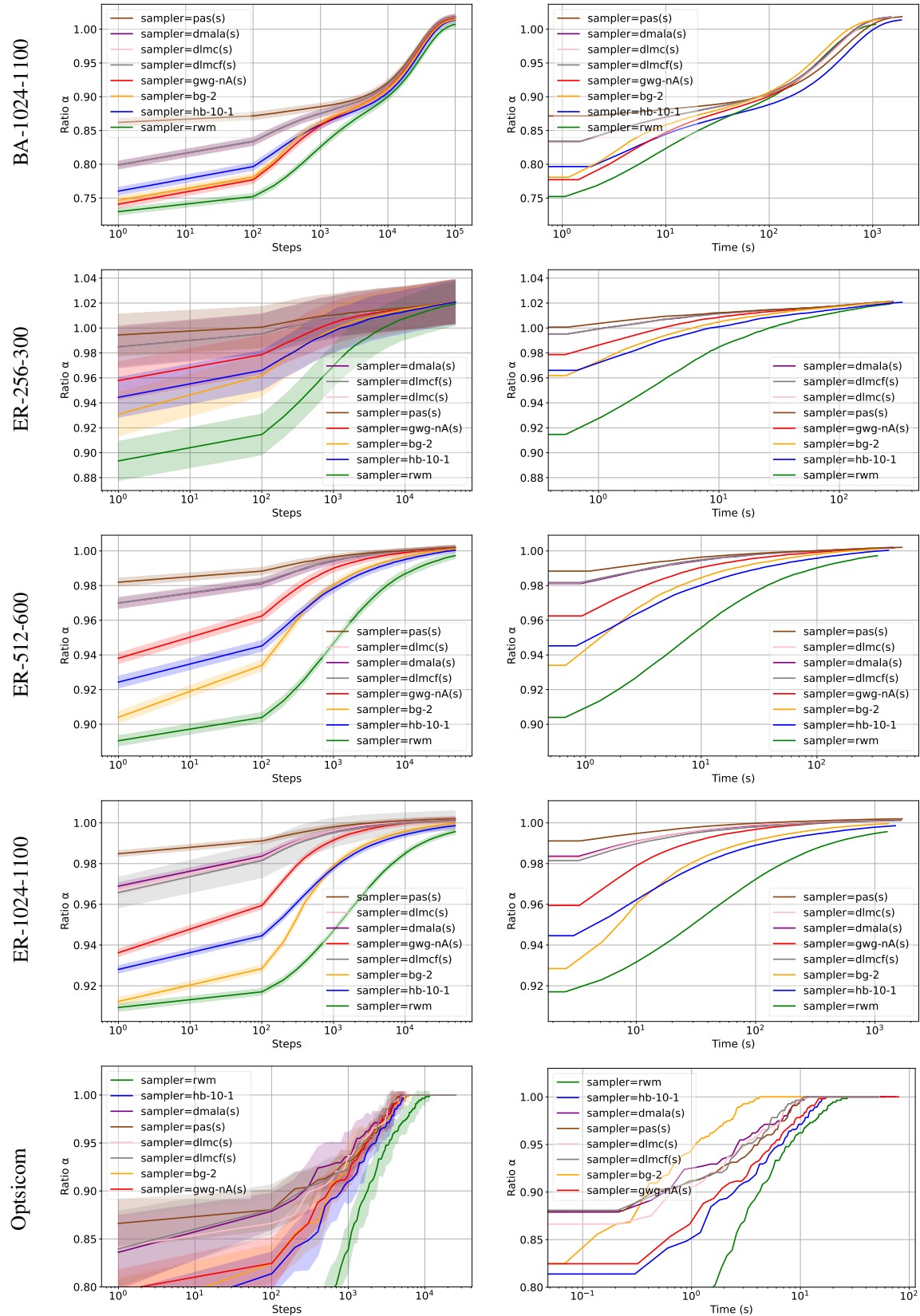

Figure 15: Solving progress on MaxCut

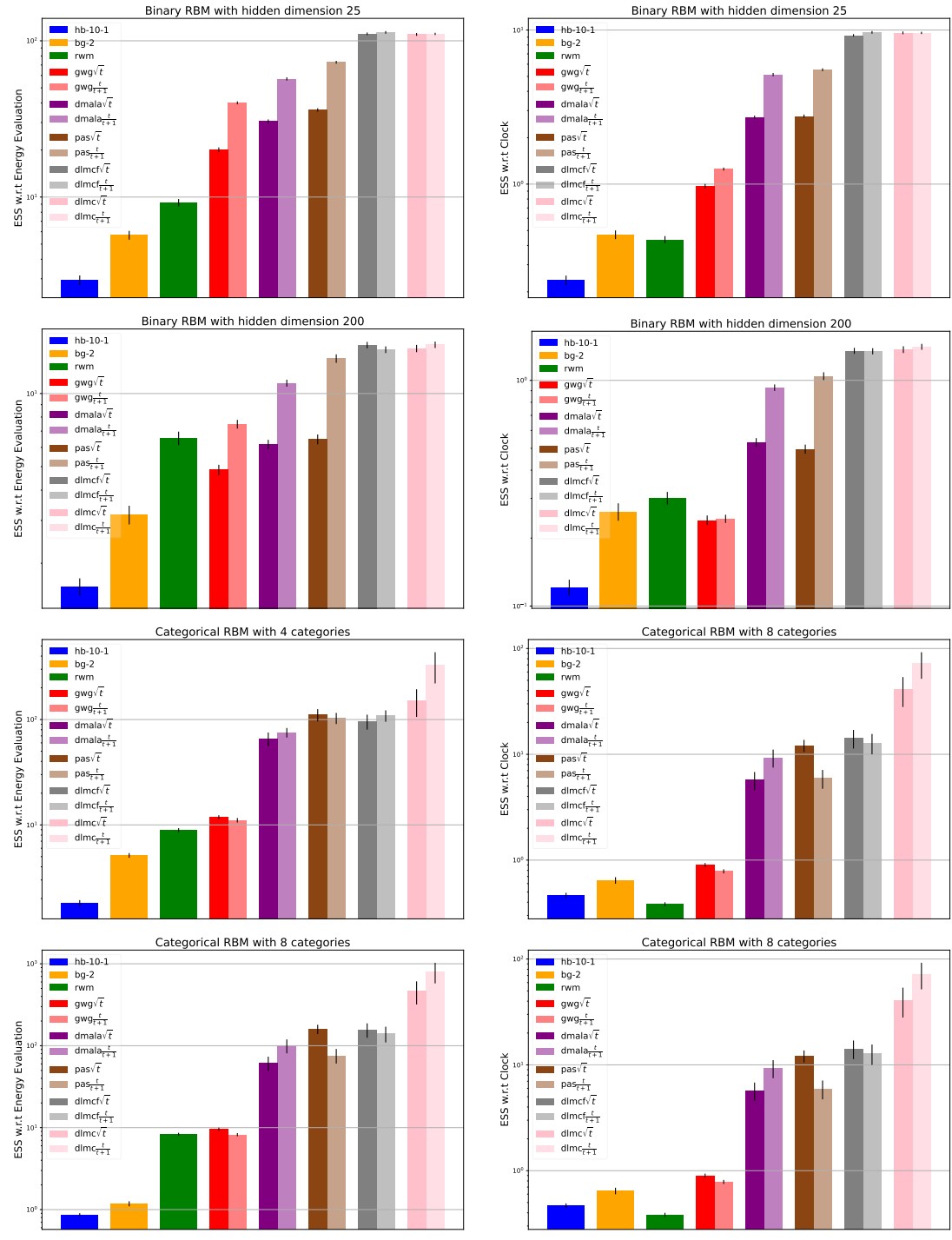

Figure 16: Results of RBMs

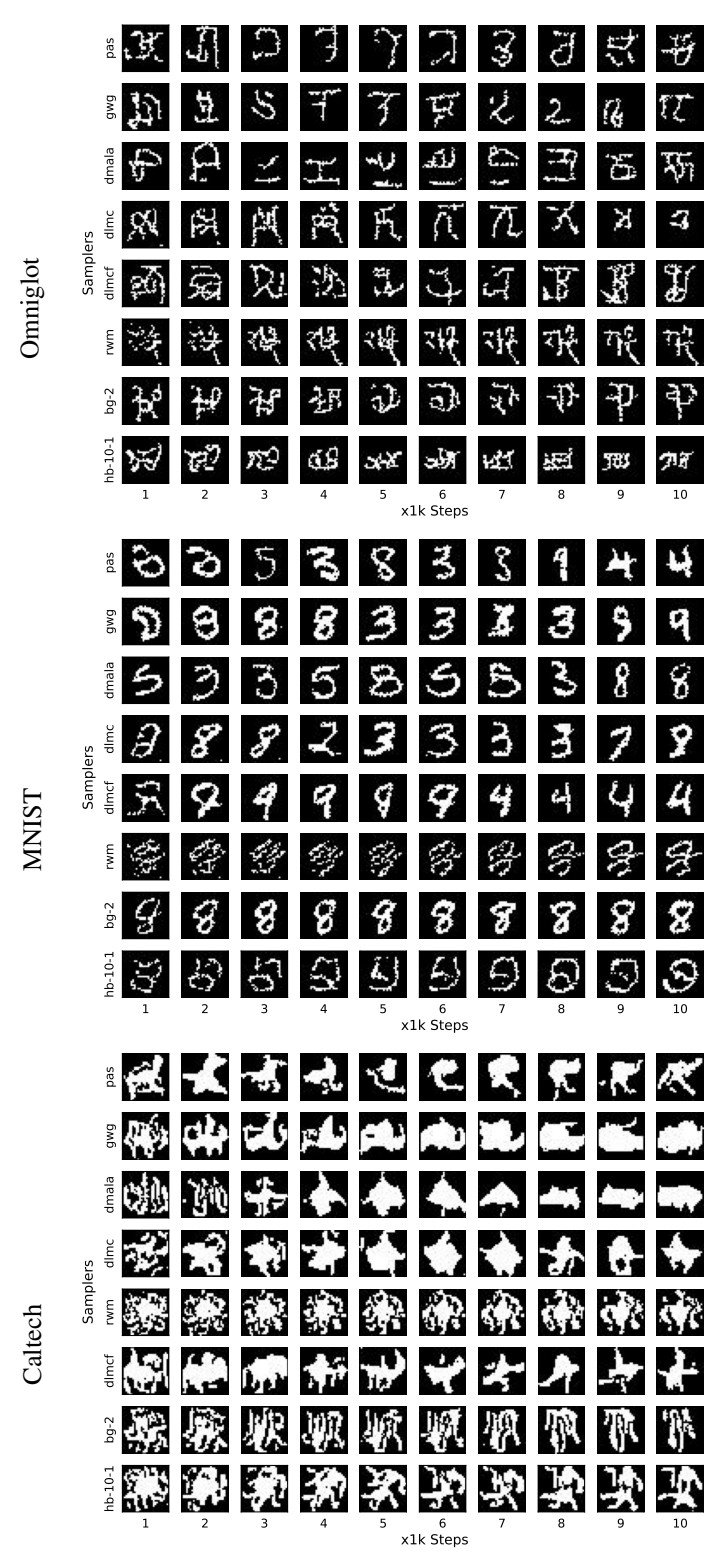

Figure 17: Resnet EBM trained on different data set with snapshots for every 1k sampling steps

