# DISCS: A Benchmark for Discrete Sampling

**Katayoon Goshvadi**
Google Deepmind

**Haoran Sun**
Georgia Tech

**Xingchao Liu**
UT Austin

**Azade Nova**
Google Deepmind

**Ruqi Zhang**
Purdue University

**Will Grathwohl**
Google Deepmind

**Dale Schuurmans**
Google Deepmind

**Hanjun Dai**
Google Deepmind

## Abstract

Sampling in discrete spaces, with critical applications in simulation and optimization, has recently been boosted by significant advances in gradient-based approaches that exploit modern accelerators like GPUs. However, two key challenges hinder the further research progress in discrete sampling. First, since there is no consensus on experimental settings, the empirical results in different research papers are often not comparable. Secondly, implementing samplers and target distributions often requires a nontrivial amount of effort in terms of calibration, parallelism, and evaluation. To tackle these challenges, we propose *DISCS* (DIS-Crete Sampling), a tailored package and benchmark that supports unified and efficient implementation and evaluations for discrete sampling in three types of tasks: sampling for classical graphical models, combinatorial optimization, and energy based generative models. Throughout the comprehensive evaluations in *DISCS*, we acquired new insights into scalability, design principles for proposal distributions, and lessons for adaptive sampling design. *DISCS* implements representative discrete samplers in existing research works as baselines, and offers a simple interface that researchers can conveniently design new discrete samplers and compare with baselines in a calibrated setup directly.

## 1 Introduction

Sampling in discrete spaces has been an important problem in physics (Edwards & Anderson, 1975; Baumgärtner et al., 2012), statistics (Robert & Casella, 2013; Carpenter et al., 2017), and computer science (LeCun et al., 2006; Wang & Cho, 2019) for decades. Since sampling from a target distribution $\pi(x) \propto \exp(-f(x))$ in a discrete space $\mathcal{X}$ is typically intractable, one usually resorts to MCMC methods(Metropolis et al., 1953; Hastings, 1970). However, except for a few algorithms such as Swedesen-Wang for the Ising model (Swendsen & Wang, 1987) and Hamze-Freitas for hierachical models (Hamze & de Freitas, 2012), which exploit special structure of the underlying problem, sampling in a general discrete space has primarily relied on Gibbs sampling, which exhibits notoriously poor efficiency in high dimensional spaces.

Recently, a family of locally balanced samplers (Zanella, 2020; Grathwohl et al., 2021; Sun et al., 2021; Zhang et al., 2022), using ratio informed proposal distributions, $\frac{\pi(y)}{\pi(x)}$, have significantly improved sampling efficiency by exploiting modern accelerators like GPUs and TPUs. From the perspective of gradient flow on the Wasserstein manifold of distributions, Gibbs sampling is simply a coordinate descent algorithm, whereas locally balanced samplers perform as full gradient descent (Sun et al., 2022a). Despite the advances in locally balanced samplers, a quantitative benchmark

Submitted to the 37th Conference on Neural Information Processing Systems (NeurIPS 2023) Track on Datasets and Benchmarks. Do not distribute.

is still missing. One important reason is that there is no consensus on the experimental setting. Particularly, the initialization of energy based generative models, random seeds used in graphical models, and the protocol of hyper-parameter tuning all have a significant impact on performance. As a result, some empirical results in different research papers may not be comparable. Under this circumstance, a unified benchmark is in crucial need for boosting the research in discrete sampling.

There are two key challenges that seriously hinder the appearance of such a benchmark. First, a sampler may perform well in one target distribution while poorly in another one. To thoroughly examine the performance of a sampler, a qualified benchmark needs to collect a set of representative distributions that covers the potential applications of a discrete sampler. Second, the evaluation of discrete samplers is complicated. Although the commonly used metric ESS (Vehtari et al., 2021) can effectively reflect the efficiency of a sampler in Monte Carlo integration or Bayesian inference, it is not very informative in scenarios when the sampler guides the search in combinatorial optimization problems, or performs as a decoder in deep generative models.

To address the two challenges, we propose *DISCS*, a tailored benchmark for discrete sampling. In particular, *DISCS* consists of three groups of tasks: sampling from classical graphical models, sampling for solving combinatorial optimization problems, and sampling from deep EBMs. These tasks cover the topics of simulation and optimization, and models ranging from hand-designed graphical models to learned deep EBMs. For each task, we collect the representative problems from both synthetic and real-world applications, for example graph partitioning for distributed computing and language model for text generation. We carefully design the evaluation metrics in *DISCS*. In sampling classical graphical models tasks, *DISCS* uses the ESS as standard. In sampling for solving combinatorial optimization tasks, *DISCS* runs simulated annealing (Kirkpatrick et al., 1983) with multiple chains and report the average of the best results in each chain. In sampling from energy based generative models, *DISCS* employs domain specific ways to measure the sample quality.

*DISCS* offers a convenient interface for researchers to implement new discrete samplers, without worrying about parallelism, experiment loop and evaluation. *DISCS* can efficiently sweep over different tasks and configurations in parallel and thus the evaluation reported in this paper can be easily reproduced. Also, *DISCS* implements existing discrete samplers

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

**Remark: scaling** Since the scalings of the proposal distribution in RWM, PAS, DMALA, and DLMC are tunable, we considers two versions with adaptive tuning or binary search tuning for fair comparison. Sun et al. (2022b, 2023b) propose adaptive tuning algorithm for PAS and DLMC when the target distribution is factorized. In practice, we find that they also apply well for other locally balanced samplers and for more general target distributions. Hence, in this paper, we use the adaptive tuning algorithm by default to tune the scaling for locally balanced samplers. In the several exceptions where the adaptive algorithm does not apply, we will use <sampler-name>-noA to indicate the results from binary search tuning.

## 4.2 Sampling from Classical Graphical Models

This section covers the classical graphical models that are widely used in physics and statistics, including Bernoulli Models, Ising Models (Ising, 1924), and Factorial Hidden Markov Models (Ghahramani & Jordan, 1995). The graphical models have large flexibility, for example, the number of discrete variables, the number of categories for each discrete variable, and the temperature of the model. The performances of different samplers can heavily depends on these configurations. *DISCS* provides tools to automatically sweep over hundreds of configurations by one click. Same as the routine in Monte Carlo integration or Bayesian inference, *DISCS* uses the Effective Sample Size (ESS) to measure the efficiency for each sampler and reports the ESS normalized by the number of calling energy function and the ESS normalized by the running time.

We use Ising Models as an example in the main text, and the more results are reported in Appendix. For an Ising Model defined on a 2D grid, where the state space $\mathcal{X} = \{-1, 1\}^{p \times p}$ represents the spins on all nodes. For each state $x \in \mathcal{X}$, the energy function is defined as:

$$f(x) = -\sum_{i,j} J_{ij} x_i x_j - \sum_i h_i x_i \tag{3}$$

where $J_{ij}$ is the internel interaction and the $h_i$ is the external field. The configurations $J$ and $h$ can be set freely in *DISCS*. In the main text, we report the results using the configuration from Zanella (2020). Specifically, $J_{ij} = 0.5$, $h_i = \mu_i + \sigma_i$, where $\sigma_i \sim \text{Uniform}(-1.5, 1.5)$ and $\mu_i = 0.5$ if node $i$ is located in a circle has the same center as the 2D grid and radius $\frac{p}{2\sqrt{2}}$, else $-0.5$. We consider the target distribution $\pi(x) \propto \exp(-\beta f(x))$, where $\beta$ is the inverse temperature. Using *DISCS*, one can easily investigate the influence of the model dimension. In Figure 1, one can see that the traditional samplers, RWM, GB, HB, have significant decrease in ESS when the model dimension increases, while the locally balanced samplers are less affected as the ratio information $\frac{\pi(y)}{\pi(x)}$ effectively guides the proposal distribution. The overall trends basically follows the prediction from Sun et al. (2022b) that the ESS is $O(d^{-1})$ for RWM and $O(d^{-\frac{1}{3}})$ for PAS.

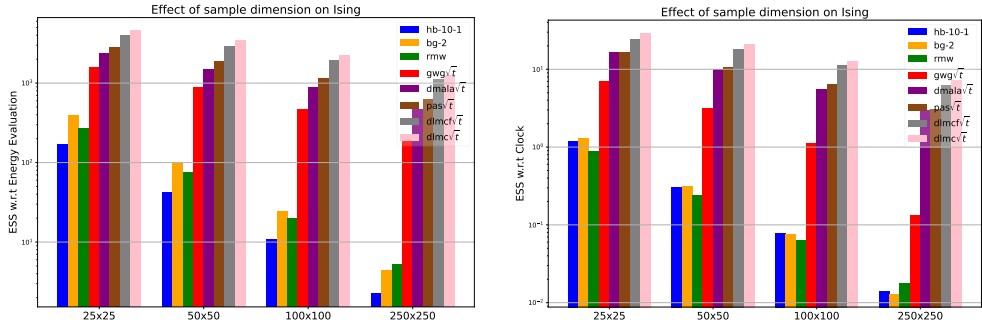

Figure 1: Results on Ising model with different dimensions

Through *DISCS*, researchers can also easily evaluate the samplers with different temperature. In Figure 2, we evaluate Ising models with inverse temperatures from 0.1607 to 0.7607. We consider Ising model without external field: $h_i \equiv 0$ and $J_{ij} \equiv 1$ as we know the critical temperature for this configuration is $\frac{2}{\log(1+\sqrt{2})}$ which means the critical point for inverse temperature $\beta = 0.4407$. From the results, we can see that

- The Ising model is harder to sample from when the inverse temperature $\beta$ is closer to the critical point, which is consistent with the theory in statistical physics
- When the inverse temperature $\beta$ is lower than the critical point, using weight function $g(t) = \sqrt{t}$ gives larger ESS; When the inverse temperature is larger than the critical point, using weight function $g(t) = \frac{t}{t+1}$ consistently obtains larger ESS.

The second observation implies that one should use ratio function $\frac{t}{t+1}$ for target distributions with sharp landscapes. We will revisit this conclusion in Figure 5 and Table 2.

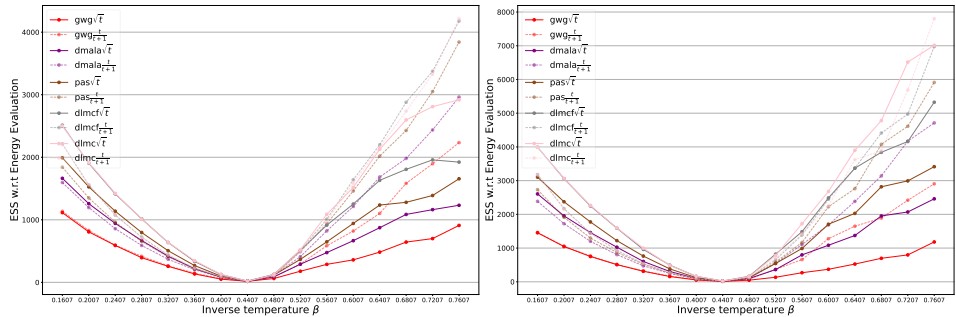

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

Table 1: Results for MIS on ER graphs. The set found by sampling algorithm is not necessary an independent set, we report a lower bound: set size - # pair of adjacent nodes in the set.

| Sampler | ER[700-800] | | | | | ER[9000-11000] |
|---|---|---|---|---|---|---|
| | 0.05 | 0.10 | 0.15 | 0.20 | 0.25 | 0.15 |
| HB-10-1 | 100.374 | 58.750 | 41.812 | 32.344 | 26.469 | 277.149 |
| BG-2 | 102.468 | 60.000 | 42.820 | 32.250 | 27.312 | 316.170 |
| RMW | 97.186 | 56.249 | 40.429 | 31.219 | 25.594 | -555.674 |
| GWG-nA | 104.812 | 62.125 | 44.383 | **34.812** | 28.187 | 367.310 |
| DMALA | 104.750 | 62.031 | 44.195 | 34.375 | 28.031 | 357.058 |
| PAS | **105.062** | **62.250** | **44.570** | 34.719 | **28.500** | **377.123** |
| DLMCf | 104.450 | 62.219 | 44.078 | 34.469 | 28.125 | 354.121 |
| DLMC | 104.844 | 62.187 | 44.273 | 34.500 | 28.281 | 355.058 |

## 4.4   Sampling from Energy Based Generative Models

The discrete samplers can also play as the decoder in generative models. In particular, given a dataset $\mathcal{D} = \{X_i\}_{i=1}^N$ sampled from the target distribution $\pi$, one can train an energy function $f_\theta(\cdot)$, such that the energy based model $\pi_\theta(\cdot) \propto \exp(-f_\theta(\cdot))$ fits the dataset $\mathcal{D}$. *DISCS* provides multiple checkpoints for the energy function trained on real-world image or language datasets. Researchers can easily evaluate their samplers after loading the learned energy function.

For the models that are relatively simple, for example, Restricted Boltzmann Machine (RBM) trained on MNIST (LeCun, 1998) and fashion-MNIST (Xiao et al., 2017b), one can continue using ESS as the metric. In Figure 5, we evaluate the samplers on RBMs trained on MNIST with 25 and 200 hidden variables. One can see that 1) DLMC has the best performance, 2) when the hidden dimension is larger, the learned distribution becomes sharper, hence $\frac{t}{t+1}$ obtains better efficiency compared to $\sqrt{t}$, which is consistent with our observation in Figure 2. For more complicated deep energy based models, a sampler may fail to mix within a reasonable steps. In this case, ESS is not a good metric. To address this problem, *DISCS* provides multiple alternative measurements, including snapshots, annealed importance sampling, and domain specific scores.

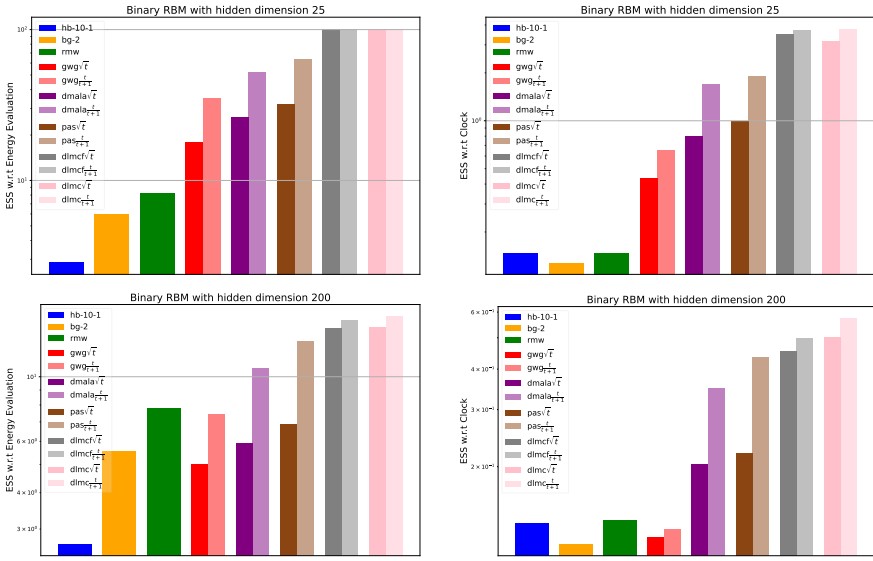

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

Of course, *DISCS* does not include all existing tasks or samplers in discrete sampling, for example, the zero order (Xiang et al., 2023) and second order (Sun et al., 2023a) approximation methods. We will keep iterating *DISCS* and more features will be added in the future. We wrap *DISCS* to a JAX library. Researchers can conveniently implement customer tasks or samplers to accelerate their study and, in the meanwhile, contribute the code to *DISCS* for further improvement. We believe *DISCS* will be a powerful tools for researchers and facilitate the future research in discrete sampling.

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

# A  Experiment Details

The source code is open source at DISCS and the data used in this paper is avaiable at DISCS DATA.

## A.1  Classical Graphical Models

For all the experiments of classical graphical models, we run 100 chains. The chains are run in parallel on 4 V100 GPUs, with each GPU handling a mini batch of 25 chains. We evaluate the performance of all the samplers and study the effect of sample shape, number of categories, locally balance function type for locally balanced samplers and the smoothness/sharpness of different models. Note that the result for BG-2 on Potts 10 and Categorical 8 model with 256 categories are omitted as it takes over 100 hours. The chain length is set as 1 million steps when studying the effect of number of categories and sample shape and in the other cases is set as 100k steps. For each experiment, as the sampling happens, all the samples of all chains are mapped separately on a randomly generated sample to a lower dimension of one. The ESS is calculated

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

Table 5: MIS.

| Sampler | Graphs | ER[700-800] | | | | | ER[9000-11000] | SATLIB |
| | Density | 0.05 | 0.10 | 0.15 | 0.20 | 0.25 | 0.15 | |
|---|---|---|---|---|---|---|---|---|
| HB-10-1 | Size | 100.374 | 58.750 | 41.812 | 32.344 | 26.469 | 277.149 | 422.427 |
| | Time(s) | 426.185 | 390.810 | 684.590 | 414.067 | 429.879 | 15139.425 | 5381.857 |
| BG-2 | Size | 102.468 | 60.000 | 42.820 | 32.250 | 27.312 | 316.170 | 422.200 |
| | Time(s) | 291.427 | 290.042 | 562.986 | 295.024 | 288.109 | 13079.125 | 3027.204 |
| RMW | Size | 97.186 | 56.249 | 40.429 | 31.219 | 25.594 | -555.674 | 420.284 |
| | Time(s) | 284.092 | 293.517 | 499.577 | 297.140 | 281.772 | 12401.737 | 2955.729 |
| GWG-nA | Size | 104.812 | 62.125 | 44.383 | 34.812 | 28.187 | 367.310 | 422.971 |
| | Time(s) | 278.885 | 308.873 | 737.671 | 303.435 | 310.551 | 24698.296 | 3540.670 |
| DMALA | Size | 104.750 | 62.031 | 44.195 | 34.375 | 28.031 | 357.058 | 423.641 |
| | Time(s) | 291.271 | 292.131 | 714.614 | 297.848 | 298.732 | 24769.380 | 3465.343 |
| PAS | Size | 105.062 | 62.250 | 44.570 | 34.719 | 28.500 | 377.123 | 424.143 |
| | Time(s) | 299.004 | 310.765 | 759.372 | 299.569 | 308.475 | 25242.166 | 4826.039 |
| DLMCF | Size | 104.450 | 62.219 | 44.078 | 34.469 | 28.125 | 354.121 | 423.387 |
| | Time(s) | 291.366 | 301.554 | 726.287 | 302.667 | 300.413 | 24892.216 | 3679.425 |
| DLMC | Size | 104.844 | 62.187 | 44.273 | 34.500 | 28.281 | 355.058 | 423.479 |
| | Time(s) | 293.235 | 294.975 | 725.326 | 294.688 | 299.884 | 24976.312 | 3523.320 |

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

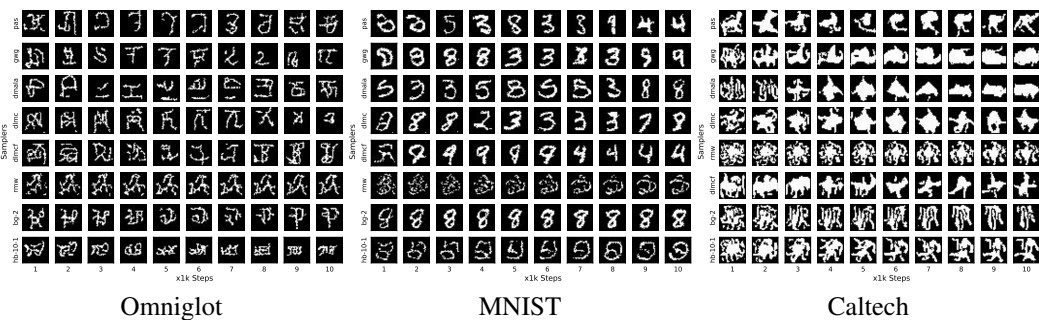

Figure 16: Results on RBMs

Omniglot       MNIST       Caltech

Figure 17: Resnet EBM