# OpenReview forum: "DISCS: A Benchmark for Discrete Sampling"
_NeurIPS.cc/2023/Track/Datasets_and_Benchmarks — NeurIPS 2023 Datasets and Benchmarks Poster_

### Official Review · Reviewer_GKfw · 2023-07-06
**A new package and benchmark of discrete sampling.**

**Rating:** 6
**Confidence:** 3
**Correctness:** Yes.
**Clarity:** Yes.

**Strengths:**

1. The discrete sampling is an important and general tool. A new convenient package and fair benchmarks would be highly valuable.
2. The new phenomenon about the local balanced weight function would provide more insights into how to select the locally balanced function.
3. The baselines are comprehensive.
4. The provided examples are comprehensive, including Classical Graphical Models, Solving Combinatorial Optimiazation and Energy Based Generative Models.


**Additional Feedback:**

See Opportunities For Improvement.

**Documentation:**

Good

**Ethics:**

No ethics concerns.

**Limitations:**

No.

**Opportunities For Improvement:**

1. The experiment results do not provide standard deviations.
2. The github document does not provide how to conveniently plug new samples into this package, thus without worrying about parallelism, experiment loop and evaluation as claimed in introduction.


**Relation To Prior Work:**

The discrete sampling is a classical question and has beed studied many years. However, early and recent works do not have consensus on experimental settings. This work provides a relatively fair comparison between these algorithms.



**Summary And Contributions:**

There is no consensus on experimental settings of discrete sampling, the empirical results in current research papers are often not comparable. And implementing samplers requires much effort in terms of calibration, parallelism, and evaluation. This paper proposes DISCS, a new package and benchmark that supports unified and efficient implementation and evaluations for discrete sampling. Throughout comprehensive evaluations, authors acquires new insights into scalability, design principles for proposal distributions, and adaptive sampling design.

---

> ### Author Response · Authors · 2023-08-18
> **Thank you for your insightful feedback. We carefully followed the reviewer’s suggestion to add more experiments.**
>
> Thank you so much for your time. Please see the details below:
>
> ### **The experiment results do not provide standard deviations**
> Thank you so much for bringing up this point. For the experiments that included reporting of the mean ESS over the number of chains, we also included the STD computed over the chains. We updated all the related plots thoroughly. For the combinatorial optimization problem, we also further updated the plots and the tables, reporting the STD of the found solutions over the instances of the model. Please find the updated plots in the revised paper.
>
> ### **The github document does not provide how to conveniently plug new samples into this package, thus without worrying about parallelism, experiment loop and evaluation as claimed in introduction.**
> Thank you so much for pointing this out. We have added more documentation explaining the details on how to plug in new samplers and define your experimental setups. The explanation could be found at:
> * https://github.com/google-research/discs/tree/main/discs/samplers

---

> > ### Comment · Reviewer_GKfw · 2023-08-29
> > **Thanks for your responses.**
> >
> > Thanks for your responses. I'd like to keep my original scores.

---

### Official Review · Reviewer_Edfj · 2023-07-18
**DISCS: A Benchmark for Discrete Sampling**

**Rating:** 7
**Confidence:** 3
**Correctness:** No concerns.
**Clarity:** The paper has a good flow and reads v…

**Strengths:**

- Discrete sampling is an important problem that is receiving more attention recently. A library was missing to quickly test various discrete samples against each other and against potentially new samplers. Thus, this is a timely and important contribution.

- The authors include a range of different discrete samples.

- It is important to note that also the methods are implemented and provided, in contrast to just providing a data set or benchmark problems.

**Additional Feedback:**

None

**Documentation:**

The benchmarks are described well, how to use the code to run the benchmarks is described well too. However, the implemented samplers could be useful outside of the specific benchmarks considered in this work and thus it should be documented how they can be used for other problems.

**Ethics:**

I don't expect any ethics concerns with this work.

**Limitations:**

- Building on JAX, the benchmark DISC can be extended with more samplers and benchmarks.

- No negative societal impact is expected.

**Opportunities For Improvement:**

The documentation of the github code mostly focuses on high-level usage but the internal functions (samplers) are little documented. It would be useful to describe how the samplers can be used outside of the specific benchmarks that are provided for other tasks.

**Relation To Prior Work:**

Adequate references are provided.

**Summary And Contributions:**

The authors provide implementations of various discrete samples together with a comprehensive benchmark set to test them.

---

> ### Author Response · Authors · 2023-08-18
> **Thank you for your insights and feedbacks. We carefully went through them and here are our responses**
>
> Thank you so much for your time and all your comments. Please see the details of our response below:
>
> ### **The documentation of the github code mostly focuses on high-level usage but the internal functions (samplers) are little documented. It would be useful to describe how the samplers can be used outside of the specific benchmarks that are provided for other tasks.**
>
> Thank you so much for pointing this out. Further explanation on how to plug in new samplers, define new experiments and how to use the samplers outside of this benchmark has been added to the documentation of the github page, following your suggestions. The explanations could be found at:
> * https://github.com/google-research/discs/tree/main/discs/samplers
> * https://github.com/google-research/discs
>
> ### **Building on JAX**
> One major reason to use JAX for the package is its ease of parallel/distributed implementation, which is the core differentiating factor of MCMC methods compared to others. We build the JAX version as the initial step, while we definitely welcome community contributions on other platforms like Pytorch as well.
>
> ### **Negative societal impact**
>
> This package is purely built as a mathematical tool for scientific usage. While it can be possible, we believe that the negative societal impact is negligible. If the reviewer has a specific list of concerns on the potential negative impact, we are more than happy to include them in the revisions.

---

> > ### Comment · Reviewer_Edfj · 2023-08-30
> >
> > Thank you for your response. I maintain my score.

---

### Official Review · Reviewer_q6pU · 2023-07-25
**Review of "DISCS: A Benchmark for Discrete Sampling"**

**Rating:** 6
**Confidence:** 3
**Correctness:** The paper and the benchmark generally…

**Strengths:**

1. DISCS covers a wide range of tasks, including sampling from graphical models, solving combinatorial optimization problems, and sampling from deep energy-based generative models, and provides a simple interface that allows researchers to implement further discrete samplers.

2. The paper introduces evaluation metrics tailored to each task type, allowing a fair comparison between different samplers.

3. Interesting observations were made through the benchmarking process, such as the difference in the performance of locally balanced weight functions as a function of the critical temperature of the target distribution.

**Additional Feedback:**

Please find further suggestions in the sections on opportunities for improvement, limitations, and clarity above.

**Clarity:**

The writing and presentation of the paper should be improved. The labels and legends in all the figures (including the ones in the appendix) are barely readable, and there are many typos (e.g., "to select locally balanced function", "we considers two versions", "by the number of calling energy function", "consider Ising model without external field", "with respect to the number of category", "consider combinatorial optimization that admit", "should depends").


**Documentation:**

The GitHub repository, in combination with the details in the appendix, provides sufficient detail to support the reproducibility of the experiments.

**Ethics:**

No.

**Limitations:**

The paper briefly acknowledges that DISCS does not encompass all existing discrete sampling tasks or samplers. However, it would be appreciated if more detail could be provided on which tasks, samplers, and metrics are omitted and why.



**Opportunities For Improvement:**

1. As already mentioned in the conclusion, DISCS does not yet compare all state-of-the-art techniques, e.g., it omits zeroth-order and second-order approximation methods.

2. The comprehensiveness of the benchmark is further limited by the fact that it does not provide extensive comparisons of different hyperparameter settings of the samplers and that only a small number of metrics is considered for some task types.


**Relation To Prior Work:**

Most of the relevant prior work is mentioned and briefly described. While additional methods could be included in DISCS, the present baselines seem to cover the most important methods.



**Summary And Contributions:**

This paper proposes DISCS (DISCrete Sampling), a benchmark and package designed to address the challenges of discrete sampling. Sampling in discrete spaces is critical in several areas, such as simulation, optimization, and generative modeling, and recent advances in gradient-based approaches using GPUs have significantly improved sampling efficiency. However, there are two major challenges that hinder progress in research: a lack of consensus on experimental settings that leads to incomparable results across different works and the non-trivial effort required to implement samplers and target distributions. DISCS aims to provide a unified and efficient implementation and evaluation framework for discrete sampling in three task types: classical graphical models, combinatorial optimization, and energy-based generative models. It provides representative discrete samplers as baselines that allow researchers to develop new samplers and directly compare them in a calibrated setup.

---

> ### Author Response · Authors · 2023-08-18
> **Thank you for your insights and feedback. We carefully went through them and here are our responses:**
>
> Thank you so much for your time and all your comments. Please see the details of our response below:
>
> ### **It would be appreciated if more detail could be provided on which tasks, samplers, and metrics are omitted and why…DISCS does not yet compare all state-of-the-art techniques…**
>
> DISCS tries to cover the most studied models and the core samplers introduced in the literature in the field of discrete sampling [1, 2, 3, 4, 5, 6, 7], by the time of the paper writing. Despite that, the samplers and tasks that DICSC currently cover might be good as the starting point to provide valuable insights for fair and efficient comparison of different methods. We want to stay humble and willing to recognize the potential missing components, and we use this to remind the readers/users to be ready for updates on baselines / datasets.
> Some recent works are not included as baselines yet, for example, the zero order [9] and second order approximation methods [9].
> With that said, our goal is to further extend the package and incrementally add more samplers and models as more papers are published, hopefully through the joint effort from the broad community.
>
> ### **it does not provide extensive comparisons of different hyperparameter…a small number of metrics is considered…**
>
> The results reported in the benchmark are gained by running the tuned hyperparameters of the samplers. In most of the experiments of the paper, we deploy the adaptive tuning algorithm [6] by default to tune the scaling for locally balanced samplers to gain the best sampler performance efficiency. In the non-adaptive cases, we studied the effect of different hyperparameters, set them as the default values and reported the results. We won’t say we have explored all the combinations of the hyperparameters, and due to the large space of tuning and limited resources, we have presented reasonable results where most of them are comparable or even better than what were reported in their original papers.
>
> To address your comments on the metrics: we have relied on the most common metrics used in the literature to evaluate the performance of sampling on different tasks. Following your advice, we added more explanations on the metrics and evaluation setup to the appendix of the paper. Our goal is to clarify the metrics used for different tasks and what they are measuring. Please refer to section A.4 in the appendix of the revised paper.
>
> We recognize that different metrics and evaluation methods could be used to measure the performance of the samplers. As a result we have dedicated a section of our paper to “Domain Specific Score”. In DISCS, we have also provided an evaluation interface for researchers to conveniently plug-in their own evaluation approach. Please find below the related documentation:
> * https://github.com/google-research/discs/tree/main/discs/evaluators
>
> ### **The writing and presentation of the paper should be improved**
>
> Thanks for this feedback. We thoroughly reviewed the paper removing typos and grammatical issues. We also updated all the plots, providing experiments standard deviations. Please refer to the revised paper.
>
> >references: \
> [1] Informed proposals for local MCMC in discrete spaces\
> [2] Oops I Took A Gradient: Scalable Sampling for Discrete Distributions\
> [3] A Langevin-like Sampler for Discrete Distributions\
> [4] LSB: Local Self-Balancing MCMC in Discrete Spaces\
> [5] Discrete Langevin Samplers via Wasserstein Gradient Flow\
> [6] Optimal Scaling for Locally Balanced Proposals in Discrete Spaces\
> [7] Revisiting Sampling for Combinatorial Optimization\
> [8] Efficient Informed Proposals for Discrete Distributions via Newton’s Series Approximation\
> [9] Any-Scale Balanced Samplers For Discrete Spaces

---

### Official Review · Reviewer_2XnF · 2023-07-28
**A Benchmark for Discrete Samplers**

**Rating:** 5
**Confidence:** 3
**Correctness:** As far as I can tell -- the main text…

**Strengths:**

Systematic frameworks are useful for head-to-head comparisons, especially when there are a number of different design choices. The benchmark is well-motivated and contains a number of existing techniques.

**Additional Feedback:**

I think that it is an interesting idea.  It isn't clear to me that the benchmark covers all (or even most) of the interesting use cases for discrete samplers.  I feel that people will still cherry-pick results...

**Clarity:**

The paper is easy to read, but contains a number of minor typos.  It would benefit from additional proofreading.

**Documentation:**

The documentation is a bit sparse:  many of the terms are undefined or taken as assumed knowledge, formulas are not given for metrics, etc.  It's probably okay for experienced researchers, but appears to require code examination if you want to go beyond the basic setup.

**Ethics:**

None.

**Limitations:**

I felt that the limitations were discussed reasonably well -- there are always more baselines to include and other tasks to consider.  The proposed tasks cover a broad range of possible use cases for samplers.

**Opportunities For Improvement:**

Given the wide variety of possible models to choose from, I'm not sure that the benchmark will achieve its intended aim of providing a robust framework for comparisons between algorithms published in different works.  Having nice baseline implementations of existing methods is useful, but I'm not sure that was the most serious bottleneck for research in practice.  The authors illustrate that the benchmarking framework can be used to identify interesting phenomena associated with critical temperatures in Ising models, but again, this could be achieved with this benchmark.

**Relation To Prior Work:**

No existing evaluation frameworks are mentioned.

**Summary And Contributions:**

The authors propose a benchmarking framework for discrete samplers.  The idea is to provide a uniform framework by which to compare existing samplers on a variety of tasks, especially given that a number of different hyperparameter settings can affect the performance of the samplers.  The benchmark contains three primary tasks:  sampling from graphical models, energy based models, and combinatorial optimization.  A number of existing baselines have already been implemented as part of the framework.

---

> ### Author Response · Authors · 2023-08-18
> **Thank you for taking the time to provide insight full feedbacks. We carefully went through them and carefully provided our responses**
>
> ### **...not sure that the benchmark will…providing a robust framework for comparisons…not sure that was the most serious bottleneck…**
> We found that the bottleneck of research in this field is a comprehensive, realistic and calibrated benchmark, as well as a calibrated implementation of the baseline methods. Sampling methods can be sensitive to the target models and initializations, so clearly a calibrated environment is the cornerstone for the measurement of the improvements. Current literature lacks a unified benchmark to achieve this goal which is our motivation to develop DISCS to tackle this bottleneck.
> To this end, we have:
> 1. Included the benchmarks from [1,2,3,4,5,6,7] while pretraining the same models (deep models, graphical models) for sampling; Included both synthetic, real-world datasets for both sampling and optimization purpose; covered multiple modalities including text, images, graphs, etc.
> 2. Implemented and tuned representative methods [1,2,3,4,5,6] to reproduce their performance, while making sure that every method is implemented in the same codebase/language/infra, to minimize the discrepancy on implementation issues. Direct and fair comparison of different sampling algorithms requires controlled experimental setup with the same target distribution, experimental setup (number of chains, chain length, etc) and evaluation metrics.
>
> ### **can be used to identify interesting phenomena associated with critical temperatures in Ising models…could be achieved with this benchmark…**
>
> DISCS’s design and interface makes the studying of the effect of different configs of models, samplers and experiments easy. We claim that its simple setup and easiness of plugging in different components, can facilitate the experimentation process, giving the convenience to use the unified set up to study different open questions and helping with gaining valuable insights.
> As an example, we provide the effect of locally balanced function type on sampler’s performance on Ising model in the paper which has been an open question for the researcher in this field for a long time.
> To be honest, we have been working in this field for a while and this is also the first time we discover these observations. We wouldn't say that this phenomena is impossible to discover without this package. Instead, we believe that a convenient package with the ease of parallel experiments would boost scientific discoveries.
>
> ### **the main text is a bit light on key details**
> Thanks for this feedback. We took this into account and revised the text to better explain the key details. Considering the broad range of samplers, models and experiments that we cover in DISCS, we focused on the main concepts and applications in the main text of DISCS and relied on the appendix for further details and explanations.
>
> ### **contains a number of minor typos**
> Thanks for this insight. We took this into account and thoroughly revised the text.
> ### **The documentation is a bit sparse**
> We took into account this insight and explained the terms and added more details in the appendix. As an example, per your suggestion, we added more details on metrics. Please refer to section A.4 in the appendix of the revised paper.
> Additionally, we added more explanation on the github page in different directories on how to conveniently plug in your own experiments, samplers, models and evaluators.
> * https://github.com/google-research/discs/tree/main
> * https://github.com/google-research/discs/tree/main/discs/samplers
> * https://github.com/google-research/discs/tree/main/discs/models
> * https://github.com/google-research/discs/tree/main/discs/evaluators
> ### **is an interesting idea. It isn't clear to me that the benchmark covers all..people will still cherry-pick results**
> The benchmark covers most of the models and tasks studied in the field of discrete space samplers [1,2,3,4,5,6,7]. From solving classical graphical models which are the most studied problems in the field to tackling more challenging problems of sampling images and texts from energy based models trained on various data. DISCS also includes combinatorial optimization problems, showing the flexibility and strength of sampling approaches.
> Also we believe this is just the version 1 of the benchmark. With the development of sampling in discrete space, we envision more challenging tasks would be included in the next versions.
> We won’t be able to prevent people from cherry-picking results, but at least a shared benchmark would make the results more reproducible and reliable.
>
> >references: \
> [1] Informed proposals for local MCMC in discrete spaces\
> [2] Oops I Took A Gradient: Scalable Sampling for Discrete Distributions\
> [3] A Langevin-like Sampler for Discrete Distributions\
> [4] LSB: Local Self-Balancing MCMC in Discrete Spaces\
> [5] Discrete Langevin Samplers via Wasserstein Gradient Flow\
> [6] Optimal Scaling for Locally Balanced Proposals in Discrete Spaces\
> [7] Revisiting Sampling for Combinatorial Optimization

---

### Decision · Program_Chairs · 2023-09-22

**Decision:**

Accept (Poster)

**Comment:**

Paper proposes a unified framework for assessment of discrete sampling under three different paraadigms.
The reviews imply that, given the objectives, the paper could be more comprehensive.
Overall, the concept and idea apper sufficiently novel to accept the paper, given that the authors intend to
further extend the framework in the future.